# Effective Application of Solid Lubricants in Spacecraft Mechanisms

**Jeffrey R. Lince** 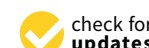

Space Tribology Consulting, Culver City, CA 90232, USA; jeff@spacetribology.com; Tel.: +1-424-218-9800

**Abstract:** Solid lubricants, antiwear coatings, and self-lubricating composites are used in applications on spacecraft where oils and greases cannot be used because of the need to avoid lubricant volatility/migration, and where the application requires significant temperature variation, accelerated testing, higher electrical conductivity, or operation in boundary conditions. The purpose of this review is to provide spacecraft designers with tools that can aid in the effective use of solid-based tribological materials, both to increase their usage, and to reduce anomalies. The various tribological material formulations are described, including how their materials, physical, and chemical properties affect their performance. Included are typical solid lubricants like PTFE and bonded or sputter-deposited $MoS_2$, as well as low shear metal coatings, hard coatings, and composite materials (including bulk composites and nanocomposite coatings). Guidance is given on how to develop mechanisms that meet performance requirements, but also how to optimize robustness, so that success is achieved even under unforeseen circumstances. Examples of successful applications are given, as well as how to avoid potential pitfalls, and what the future of solid tribological materials may hold.

**Keywords:** solid lubrication; antiwear coatings; spacecraft; tribology; molybdenum disulfide; nanocomposites; ball bearings; actuators; slip rings; deployment mechanisms

## 1. Introduction

Along with oils and greases, solid lubricants and anti-wear coatings have been used successfully since the early days of the space program [1,2]. Solid-based materials are especially useful when used in applications with large temperature extremes or variations, such as cryogenic sensors [3] and turbopumps for LH2/LOX fueled engines [4]. They are also useful where material containment is an issue because the lubricant must be kept in the contact region or to avoid migration onto nearby sensitive optical components [5]. There are also applications for which either solid or liquid lubricants can be

used, with the choice between them driven by more subtle requirements; such applications include slip ring assemblies [6,7], actuators [8], gimbal bearings [9], and even reaction wheels (RWs) [10].

In general, optimizing tribology in the space environment is difficult for several reasons. First is the vacuum environment, necessitating using materials with very low vapor pressures. This has limited liquid-based lubricants to those based on PFPE, MAC, and PAO oils [11]. However, this concern is mostly moot for solid lubricants: in typical spacecraft operating environments, they exhibit negligible vapor pressure. A second challenge is the remoteness of the application: once launched, there is usually no opportunity to replace the lubricant or service the lubricated device. As such, long and unattended life is a requirement. This implies that the solid lubricant must exhibit low wear rate, since replenishment of solids on orbit is difficult, although composites can be used in some applications to provide a reservoir of solid lubricant material.

Third, tribological materials often must be able to be operated in air as well as in the vacuum of space, because they must be tested prior to launch. This can prove challenging. For example, $MoS_2$ is an excellent solid lubricant used extensively in spacecraft applications because of its ability to lubricate effectively in vacuum and inert environments, but performs somewhat poorer when operated in air or when stored in humid air environments before launch [2,12–15].

There are relatively few types of solid lubricants used on spacecraft. Molybdenum disulfide ($MoS_2$) is the most commonly used [15], while PTFE and metals with low shear strength such as lead (Pb) are used in niche applications like ball bearings [2,5,11,16]. There has also been limited use of hard coatings tailored to exhibit reduced friction, such as TiCN [17].

As ubiquitous and useful as low friction/wear solids have proven to be, it has not been all smooth sailing. Because of a lack of understanding of their basic chemical and materials properties, solid lubricants are often used in applications for which they are not appropriate. For example, PTFE is an excellent solid lubricant with a low coefficient of friction (COF), but its low shear strength limits its use to low loads.

The picture becomes more complicated because although there are only a few solid lubricating/antiwear materials used in spacecraft, there are many ways to formulate these materials for actual use. For example, $MoS_2$ powder can be mixed with curable epoxies to form at least several hundred types of bonded coatings, $MoS_2$ can be codeposited with other materials by Physical Vapor Deposition (PVD) to form thin adherent coatings, or $MoS_2$ powder can be burnished or applied using high pressure air to form thin layers. Each formulation has a subset of spacecraft mechanism applications for which it is optimum, and others for which it is not appropriate.

Finally, no solid lubricant is a panacea, so it is important to put effort into enhancing the robustness (resiliency in the presence of unplanned adverse conditions) of a tribological contact. This means not only choosing the best solid lubricant formulation for the application, but also choosing substrate materials correctly, and/or including additional coatings such as hard materials on the surfaces prior to applying lubricants.

This paper will (1) describe what types of low friction solids are available for spacecraft use, (2) provide guidance on the best formulations for different applications, (3) and show how understanding chemical and materials properties of these formulations may guide their successful use. It is important to evaluate the use of solid lubricants and antiwear coatings early in the design phase of mechanisms; choosing and applying a lubricant coating after the mechanism is designed (and sometimes even after it is built) can result in poor robustness and even device failure.

Solid lubricants (as thin coatings or powders) are typically used for:

- Low to medium numbers of duty cycles
- Moderately-high to low contact stresses
- Extreme environments
- Low speed boundary contacts

Fluid lubricants (oils and greases) are used in mechanisms with many cycles, either rotational (e.g., ball crossings in a ball bearing) or reciprocal sliding, where contact stresses are determined by the yield-strength properties of the load-bearing materials, and where the environment is relatively benign. They perform best at higher speeds in the elastohydrodynamic (EHD) regime. In most cases of fluid lubricant use, the apparatus is engineered to confine the lubricant so that it is not lost from the critical contact regions by means of evaporation, creep, or centrifugal motion. For solid lubricants it is also necessary to confine the lubricant in the contact region, but this task is easier because the physical mechanisms acting on them are different from fluids. Several authors have discussed the advantages and disadvantages of solid versus fluid lubricants, [11,18,19] and these are summarized in Table 1.

**Table 1.** Comparison of Solid and Fluid Lubricants for Space Applications.

| Property | Solid Lubricants | Liquid Lubricants |
|---|---|---|
| Volatility/Migration | Negligible volatility/Debris migration possible | Finite vapor pressure/Confinement required |
| Endurance | Life determined by lube wear—resupply difficult | Resupply possible—longer life |
| Temperature Range | Wide operating temperatures | Physical properties vary significantly with temperature |
| Friction/Torque Noise | Friction sensitive to debris → torque noise | Low torque noise—good models for torque calculation |
| Possibility of Accelerated Testing | Accelerated testing possible if failure mechanism known | Accelerated testing difficult |
| Air/Moisture Sensitivity | Life/friction can be air/moisture sensitive | Less sensitive to air/moisture |
| Electrical Conductivity | Electrically conducting | Poor electrical conductors |
| Substrate Sensitivity | Good adhesion is critical | Additives need to be matched to substrate |

There are numerous excellent works on the subject of solid lubrication that are not covered here; the emphasis in this work is on spacecraft applications, which greatly limits the scope of discussion.

## 2. Material Properties, Structure, and Lubrication Mechanisms

Solid lubricants—including lamellar solids, polymers, metal salts, and soft metals—all have specific attributes that make them good choices for particular applications. However, solids generally possess the ability to operate in extreme temperature environments with little or no contamination of surrounding critical surfaces (e.g., optics and thermal control surfaces). In addition, they are unaffected by storage for long periods, although they can be affected by storage in humid air, depending on the formulation. General properties that make a good solid lubricant are low shear strength, good adhesion to the surfaces to be lubricated, low abrasivity (i.e., they must be softer than the substrates), and thermodynamic stability in the application environment [20].

Since lubricating materials often consist of mixtures of species (e.g., bonded coatings and bearing cage materials), materials/engineering properties of the individual components may have little relation to the composite lubricant. However, some visibility into their expected properties may be obtained

from standard reference texts. For example, Ref. [21] provides information on a variety of polymers, ceramics, steels, and other metal alloys.

Coatings derived from the lamellar transition metal dichalcogenide (TMD) $MoS_2$ and those consisting of polymers such as PTFE are commonly used on spacecraft. In some cases, these two materials are combined to form self-lubricating bushings or ball-bearing cages. Both materials are characterized by very low sliding friction coefficients, typically in the 0.02–0.1 range, with values as low as ~0.002 for sputter-deposited $MoS_2$ coatings under optimum preparation and running conditions [22]. In general, the reasons both materials lubricate well are similar: they provide very low energy, repulsive surfaces on opposing substrates that result in minimum dissipation of energy when the substrates rub against each other. However, the chemical structures, and hence the molecular origins of low friction, of the two materials are quite different.

### 2.1. Transition Metal Dichalcogenides—$MoS_2$ and $WS_2$

TMDs like $MoS_2$ and $WS_2$ are lamellar solids consisting of layers of material with strong chemical bonds within the layers but weak physical-type (van der Waals) bonding between layers (see Figure 1a) [15]. Low friction is a direct result of the weak van der Waals bonding, allowing layers to readily slide over one another. The initial lubrication process involves shear of the TMD materials and movement of many crystallites within the area of contact, with some transfer of particles from surface to surface. After a run-in period, at steady state, the movement of crystallites is more limited, with most sliding occurring between the coated surface and the opposing surface on which a stable transfer film has deposited. The shear process combined with the anisotropic crystal structure of TMD's results in the plate-shaped crystallites forming at the surface with their basal planes aligned with the sliding direction [15].

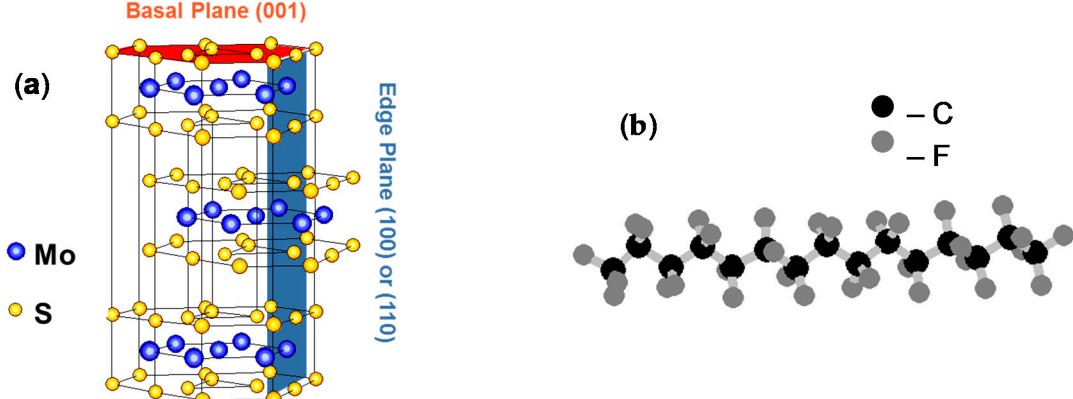

**Figure 1.** Molecular origins of low friction in lubricants based on (**a**) $MoS_2$ and (**b**) PTFE. Both structures rely on reduced dimensionality for low friction. $MoS_2$ exists in a pseudo-two-dimensional structure consisting of stacked S-Mo-S "sandwich" layers. There is strong covalent bonding within the sandwich layers, but relatively weak van der Waals bonding between these layers. Lubrication occurs because crystallites in contact with one another are incommensurate, i.e., the (100) and (110) crystallographic directions are different for the two crystallites, while their (001) directions are in approximate alignment. Lubrication with PTFE is similar in that pseudo-one-dimensional chains exhibit strong covalent bonding within a chain, and weak van der Waals bonding between either other chains or a contacting surface.

Although the basal plane surfaces of the crystallites are aligned in the contact, low friction is only achieved when there is a lack of rotational registry, that is, the atoms of one tiny crystallite platelet do not line up with those of adjacent platelets (also known as incommensurability) and some repulsion between the platelets results [22,23]. It is believed that the mechanism behind this reduction in friction is that the lateral force experienced by atoms during sliding are canceled out by the lateral force in the opposite direction associated with another atom, thereby resulting in negligible net friction force. For materials with small crystallite sizes, including sputter-deposited $MoS_2$ coatings (see Section 3.4), as the degree of incommensurability increases, the COF correspondingly decreases, reaching values lower than 0.001, which is known as superlubricity [24]. This was demonstrated by direct measurement of the COF between a single-layer flake of $MoS_2$ and a bulk $MoS_2$ substrate, using a silicon nanowire-based mechanical force sensor. The interaction was observed inside a scanning electron microscope (SEM), which confirmed incommensurate registry between the two. In this idealized contact, a COF as low as 0.0001 was observed [25].

TMDs like $MoS_2$ behave according to the Hertzian Contact Model [26], where the coefficient of friction (COF) decreases as the normal load is increased. Specifically, the COF for $MoS_2$ is inversely proportional to the Hertzian contact stress, unlike solids that follow Amontons' Law, where the COF is independent of load [27]. This behavior is caused by the low shear strength of TMD's, as opposed to harder materials where the COF is related to surface interaction.

The shear strength of a particular TMD is related to the ability of crystallites to move relative to each other. As such, adhesion between crystallites would be expected to vary with differences in crystallinity. Bulk $MoS_2$ exhibits a shear strength of 38 MPa [26]. Pure sputter-deposited $MoS_2$ exhibits lower values for typical coatings (7 MPa) [28]. Pure $MoS_2$ coatings with high sulfur vacancy defect levels exhibit higher values (40 MPa) [29], and nanocomposite amorphous Au-$MoS_2$ coatings higher still (83 MPa) [30]. As will be discussed in Section 6.2, shear strength depends on sliding environment also.

For coatings with poor crystallinity or even a completely amorphous TMD phase, low friction is rapidly achieved during initial sliding, as demonstrated in Figure 6 and by related discussion in Section 3.4.

The relative performance of different TMD's is primarily related to subtle differences in their crystal and electronic structures. For example, $WS_2$ is a good lubricant; it has virtually the same structure as $MoS_2$ because W is in the same group as Mo in the periodic table. Similarly, $MoSe_2$ is a good lubricant, since S is in the same group as Se. However, $MoS_2$ has proven consistently superior at typical operating temperatures in vacuum than $WS_2$ [31,32] and $MoSe_2$. This superiority may be due to the less diffuse electron orbitals on the smaller Mo and S atoms, resulting in a surface with poorer bonding properties. Such poorer bonding gives rise to lower shear force and adhesion, with resultant low friction.

## 2.2. PTFE

PTFE is a crystalline polymer material consisting of arrays of long "zigzag" chains or helixes that are held together by primarily physical forces. The bonding within the chains is covalent and strong, while that between chains is weaker, providing a one-dimensional analog to the two-dimensional lamellar TMDs (see Figure 1b). Fluorine atoms are bonded to the "backbone" carbon atoms, causing the molecular chains to twist into helixes. Compared to hydrogen atoms in polyethylene, fluorine atoms are highly electronegative and bulky. Like the layers in $MoS_2$, these helixes do not easily form chemical bonds to other atoms or molecules, resulting in a low-energy, low-friction surface. (This is demonstrated by the beading of water on PTFE-coated cookware.)

The lubricating action of PTFE results from alignment of the molecular chains in the direction of motion and drawing out of chains into the contact region [33–35]. Formation of uniform transfer films of aligned molecular chains occurs under conditions of low energy dissipation (i.e., slow speed, low loads). The alignment of molecular chains to form fibrillar structures of PTFE on an uncoated glass

surface was demonstrated using SEM and Scanning Force Microscopy (SFM) [36]. The orientation was shown to occur during shearing of the transfer film that sticks glass surface due to the high adhesion between the first layer of PTFE and the glass. The ordered molecular films provide low, uniform friction (low noise) within the contact.

At loads higher than the yield strength, as well as higher speeds that result in high contact temperatures, uneven transfer of what are essentially melted lumps of polymer results; this causes nonuniform (i.e., noisy) friction behavior until the films are spread thin. Such films can be smoothed out with continued operation only if the temperature in the contact region is lowered, which is typically unlikely.

*2.3. Surface Reaction Layers*

Mechanical devices made from soft or easily oxidized metals like aluminum and titanium alloys can be made more robust by the formation of surface reaction layers. Chemicals are reacted with the surface of the metal, to produce, for example, a metal oxide or metal phosphate layer. The most common process for spacecraft mechanical components is anodization. For aluminum and titanium, the anodization processes conform to SAE AMS2469 [37] and SAE AMS 2488 [38], respectively. Reaction layers prevent corrosion by inhibiting oxygen diffusion to the underlying metal surface. The formation of reaction layers on Al and Ti is usually required before application of bonded solid lubricant coatings on their surfaces. (Although the bonded coatings provide a small amount of corrosion protection, it is usually not adequate.)

Tribologically, there are three reasons for using such reaction layers. First, the oxidized form of the layer is in a more chemically stable state than the underlying metal. The layer prevents metal-to-metal contact, which can result in adhesive wear and subsequent galling, as discussed in Section 4. Second, the layers are harder than the underlying metal, resulting in a significantly lower wear rate [39]. Finally, the reaction layer can lessen deformation of the underlying soft metal surface; such deformation could cause a solid lubricant coating deposited on the part's surface to weaken or debond.

## 3. Application and Use of Solid Lubricant Formulations

The choice of correct formulation for a specific application depends on the expected performance properties. For spacecraft lubrication systems, critical parameters are coefficient of friction, load carrying capacity, electrical conductivity, temperature, and presence of liquid oxygen or radiation. Environmental exposure during terrestrial testing and storage should also be considered, including resistance to degradation from oxygen, humidity, and other atmospheric gases.

General guidance as to the relative properties of different solid lubricant formulations is listed in Table 2. Information in this table can help to understand the tradeoffs between competing performance characteristics. However, there are exceptions to these general guidelines. For example, the values for the mean Hertzian contact stress limit ($S_m$) listed in Table 2 are typical for many applications, but they could vary by as much as a factor of 2 depending on the type of contact (sliding versus rolling) and the number of cycles required. In addition, $S_m$ may vary during different regimes of operation, and so could be considerably higher than the average much of the time. (A more thorough discussion of Hertzian contact stress, including online stress calculators, is presented in Ref. [40].)

**Table 2.** Design Guidelines for Solid Lubricants and Antiwear Materials.

| Lubricant Coating Formulation [a] | Property | | | | | | | | | | | |
|---|---|---|---|---|---|---|---|---|---|---|---|---|
| | Mean Hertzian Contact Stress Limit ($S_m$, ksi) [b] | Friction, in Vacuum | Approx. Thick-Ness | Thickness Variation | Min. Temp. (°C) | Max. Temp. (°C) | Adhesion | LOX Compatibility | Sliding/Rolling (S/R) [c] | Relative Endurance | Robustness to Humid Air Storage | Resistance to Condensed Moisture |
| Unbonded $MoS_2$ or $WS_2$ | 200 | 0.02–0.1 | 0.1–10 μm | ±80% | −260 | 900 (in vacuum) | Low | Yes | S/(R) | Very Low | Fair | Fair |
| Unbonded PTFE | 2 | 0.02–0.2 | 1 μm | ±80% | −35 | 150 to 250 | Low | Yes | S/(R) | Low | Very good | Very good |
| Resin-bonded/Heat Cured | 100 | 0.03–0.1 | 10 μm | ±50% | −220 to −70 | 200 to 400 | Medium | No | S/(R) | High | Very good | Good |
| Resin-bonded/Air Cured | 50 | 0.03–0.1 | 10 μm | ±50% | −220 to −70 | 150 to 400 | Medium | No | S/(R) | Medium | Good | Fair |
| Inorganic-bonded | 150 | 0.03–0.2 | 10 μm | ±50% | −250 to −70 | 370 to 850 | Medium | Yes | S/(R) | Medium | Fair | Poor |
| Ceramic-bonded | 100 | 0.1–0.2 | 10 μm | ±50% | −240 to 20 | 600 to 1100 | Medium | Yes | S/(R) | Medium | Fair | Poor |
| Sputter-Deposited $MoS_2$ | 200 | 0.003–0.05 | 1 μm | ±10% | −260 | 400 | High | Yes | S/R | High | Fair/Good [e] | Fair/Good [e] |
| Ion Plated Pb | 130 | 0.1–0.3 | 0.1–1 μm | ±10% | −260 | 300 | High | Yes | (S)/R | Medium | Fair | Fair |
| Carbide/Nitride Coatings | >300 | 0.4–0.8 | 0.1–2 μm | ±10% | Varies | >700 | High | Yes | (S)/R | High | Very good | Very good |
| HH-DLC | >200 | 0.001–0.01 | 1 μm | ±10% | < −200 | 300 | High | Yes | S/R | High | Very good | Very good |
| Bulk Lubricating Materials [d] | 1–50 | 0.02–0.4 | ~1 cm | N/A | Varies | Varies | N/A | Yes | S/R | High | Varies | Varies |

[a] Formulations are often combined to optimize properties, e.g., using both sputter-deposited $MoS_2$ coatings and a PTFE-containing cages in ball bearings. [b] Actual limit to mean contact stress will vary with, e.g., type of application, sliding/rolling speed, and expected lifetime, etc. [c] Minor application is listed in parentheses (). [d] Properties vary strongly with specific material and application; e.g., pure PTFE journal bearings versus composite PTFE-based bearing cages. [e] Improved by cosputtering with materials such as metals or $Sb_2O_3$, or by forming metal/$MoS_2$ multilayers.

Choosing the best solid lubricant formulation for an application is just the first step. Care should be taken to apply the coating correctly, since the structure and tribological properties of a given coating can vary greatly. Failures have occurred in applications where the coating preparation procedure was changed only slightly from a previously successful application. Optimum preparation includes preparation of the part surface: roughness and cleanliness affect both adhesion and wear properties.

Successful application of tribological solids and coatings is often aided by consultation with personnel at lubricant manufacturing companies. There is also extensive industrial experience available in the literature (see, for example Ref. [41]). Finally, testing (discussed below) should be designed to simulate the varying stresses and other conditions seen in the actual space mechanism to evaluate if the lubricant is sufficiently robust for use.

### 3.1. Surface Pretreatment for Thin Lubricant Coatings

The surface(s) of the part or mechanism must be appropriately prepared before coating application, primarily to increase the subsequent adhesion of the coating. Surface cleanliness is necessary for all coating growth techniques, since organic surface contamination can prevent strong bonding between coating and substrate.

Initial cleaning usually involves treatment with organic solvents and/or caustic cleaners. For example, for heat-cured, bonded solid lubricant coatings, Ref. [42] specifies precleaning the surfaces with an aliphatic hydrocarbon solvent conforming to ASTM D3735, or any environmentally safe method that cleans surfaces to meet the requirements of ASTM F22, and does not harm the surface.

Depending on the type of lubricant coating, additional pretreatment steps may be necessary. This is especially important for bonded coatings, which require surface roughening and either plating or passivation of the part surface, as discussed below. Even unbonded coatings (e.g., burnished $MoS_2$ powder) perform better with some surface roughness [1,2,43]. However, it should be understood that a plot of endurance versus RMS surface roughness has a maximum at a different position for applications with differing part material, coating type, and even roughening method (discussed below). As such, the optimum surface roughness must be found empirically for any nonstandard application. However, there is an upper bound to acceptable surface roughness: a rule of thumb is that the value for $R_a$ should be less than half the coating thickness.

### 3.2. Unbonded/Burnished Coatings

Unbonded lubricant coatings are usually fabricated from $MoS_2$, $WS_2$ or PTFE powders. One of the earliest ways to apply solid lubricants was to burnish the lubricant powder onto a part's surface using a brush or cloth. The burnishing compacts the coating and enhances the surface area of the substrate covered with coating. In the case of $MoS_2$ or $WS_2$, burnishing also causes orientation of the crystallites within the coating such that the basal planes are mostly parallel to the sliding direction, allowing low friction (see Figure 2). The powders are bonded to the substrate surface by relatively weak van der Waals forces, which limits the adhesion.

The powder can also be mixed with a volatile solvent for spraying, brushing, or dipping onto the part's surface: the solvent evaporates, leaving the pure powder.

Although application of unbonded coatings is simple and inexpensive, the resultant coating is poorly adherent. Also, they can exhibit poor reproducibility, with variations in coating thickness and particle morphology. The thickness depends on several parameters that are difficult to control. For example, the thickness of burnished $MoS_2$ coatings may vary from 0.1 to 10 μm (4 to 400 μin), depending on burnishing time and method (e.g., cloth versus tissue), surface roughness, and even the relative humidity during burnishing [44]. However, even if thickness can be controlled, after a short amount of device operation, an unreproducible amount of $MoS_2$ is likely to be removed from the contact region, leaving a coating with unpredictable, and usually much smaller, thickness.

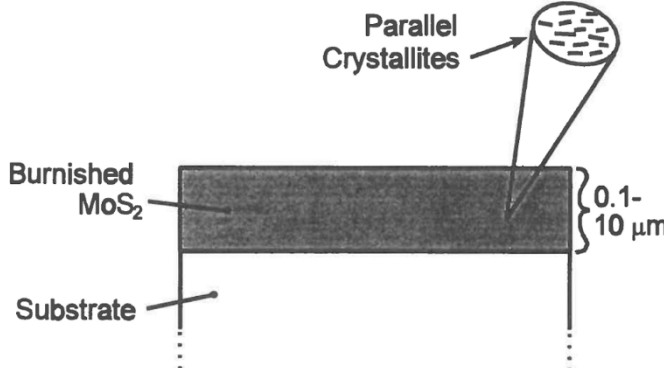

**Figure 2.** Schematic cross section of a MoS$_2$ coating produced by burnishing pure MoS$_2$ or WS$_2$ powder onto a surface. Similar structures are produced by air-impingement of the powder, as discussed in Section 3.2.1.

Such variations and the poor adherence of unbonded coatings limit use to noncritical or undemanding applications. For example, burnished MoS$_2$ and PTFE may be used as antiseize lubricants during installation of screws, rivets, and connectors on space hardware. Attempts to use unbonded coatings as the sole lubricant in bearings, even for low numbers of cycles, have met with failure.

However, burnished MoS$_2$ coatings can provide improved bearing performance when used in combination with self-lubricating cages (containing MoS$_2$ and/or PTFE). This enables a more graceful run-in, with lower initial wear. Although sputter-deposited MoS$_2$ coatings have been demonstrated to perform this task successfully, sputter-deposition may be impractical if, for example, the bearing size is too large.

3.2.1. Special Case: Air-impinged MoS$_2$ and WS$_2$

Improved adhesion of unbonded coatings can be achieved by spraying MoS$_2$ and WS$_2$ in high pressure air to form a thin film on a surface, akin to a "grit-blasting" process. The most common version of this process is Dicronite DL-5$^{TM}$, which licenses vendors around the world to apply it. Other vendors apply WS$_2$ using similar processes. Its advantages include ease of applying to multiple surfaces of a part, low cost, thickness (which is typically <1 μm), and relatively low COF value (similar to MoS$_2$). The improved adhesion is due to a mechanical process whereby the impinged soft lubricant particles deform to match features on the surface. Surface preparation generally involves grit-blasting abrasive powder (e.g., aluminum oxide) onto the surface to clean and roughen it. (Care must be taken to remove any residual abrasive prior to applying the solid lubricant.)

As for other nonbonded coatings, the thickness and coating structure can depend on the environment, especially humidity (higher humidities can give rise to thicker films).

Because of the relatively low adhesion of air-impinged coatings, sputter-deposited and bonded MoS$_2$ and WS$_2$ coatings show significantly higher cycle life. A study was conducted by ESTL personnel comparing Dicronite coating performance from different vendors to sputter-deposited MoS$_2$ [31]. Pin-on-disk (POD) testing of the coatings was conducted in nitrogen. The endurance of a sputter-deposited MoS$_2$ coating was found to be two to three orders of magnitude greater than the Dicronite coatings.

In that same study, a ball bearing pair was coated with Dicronite (rings, balls, and steel cages) and tested in vacuum. The COF was just below 0.1 after a run-in period, and the bearing pair failed at 112,000 revs. In comparison, similar bearings tested by ESTL coated with sputter-deposited MoS$_2$ showed COF values in the range 0.02 to 0.05, and endurance values up to 3 million revs [31].

Another study evaluated the use of Dicronite in harmonic drive bearings [32]. The bearing tested was a hybrid using Si$_3$N$_4$ balls and a cage made from a self-lubricating polyimide-MoS$_2$ composite.

Dicronite was applied to the race surfaces to act as initial lubrication during run-in, until $MoS_2$ from the cage could be transferred to the balls and races. They found the gear efficiency to be low, and ascribed this to high friction in the bearings due to poor performance by the Dicronite. The Dicronite wore prematurely, as evidenced by $WS_2$ wear debris that were seen in SEM/EDX analysis of the gear teeth. As a result, the Dicronite was replaced with a sputtered $MoS_2$-based coating in the application.

More than other types of coatings, the key for air-impinged coatings is finding a vendor with excellent quality control. Without adequate care, the coatings' cycle life is unpredictable. In illustration, the study in Ref. [31] found that endurance of Dicronite coatings varied by two orders of magnitude among three different vendors.

The strengths of this type of coating are that the coatings are thin and relatively easy and inexpensive to apply in large quantities. As such, they have found successful application in a variety of uses requiring relatively few numbers of cycles. In spacecraft, they have been useful as antiseize lubricants for fasteners. In addition, they have found niche usage in low cycle mechanisms including ball bearings.

*3.3. Bonded Coatings*

Bonded coatings consist predominantly of two components, the lubricant and the binder. The lubricant is generally a powder (with particles 1–10 μm in diameter) that is dispersed within the binder material. The binder provides adhesion of the lubricant to the surface of the part and controlled wear of the lubricant coating due to its higher hardness and strength. Bonded coatings can be classified in terms of the type of binder, types of lubricants/additives, and the curing method [1,19,43,45,46]. There are hundreds (and perhaps thousands) of bonded solid lubricant products available, with a wide range of lubricants, binder materials, and additives. Current binder materials for aerospace applications include thermoplastic and thermosetting resins for lower temperatures, phosphates and silicates for low to moderately high temperatures, and ceramics for higher temperatures.

Figure 3 shows the cross-sectional structure of a bonded coating. When the coating is applied to a mechanical device, the lubricant particles are spread uniformly throughout the binder. However, when properly burnished/run-in, the surface of the coating should consist almost entirely of the lubricant. In the case of bonded $MoS_2$ coatings, the surface layer consists of $MoS_2$ crystallites with their basal or (0001) crystallographic planes aligned along the top surface of the coating. (Such surface enhancement/alignment occurs for all $MoS_2$-based composite materials and coatings, and is demonstrated for nanocomposite Au-$MoS_2$ coatings in Section 3.4) In addition, the binder allows controlled wear of the coating. The wear must be low enough to provide lubrication over the lifetime of the part, but must be high enough to allow continuous replenishment of the lubricant to the sliding surfaces of the parts.

Bonded coatings for space are mostly based on $MoS_2$ and to a lesser extent PTFE. Graphite is a common additive for terrestrial applications, but not generally appropriate for spacecraft, especially if used alone (see Section 6.4). Additional materials can be added to provide improved properties; examples include $Sb_2O_3$, PbO, and other proprietary formulations.

For resin-bonded $MoS_2$ coatings, the optimum lubricant-to-binder ratio is in the range from 1:1 to 4:1, with 2:1 being typical [43,46]. For air-impinged $MoS_2$ coatings, smaller amounts of a binder containing sodium silicate (or other inorganic species), with a lubricant-to-binder ratio of 20:1 have been used. Higher lubricant amounts give reduced friction, but higher binder amounts give greater wear life, corrosion resistance, and hardness, with higher friction.

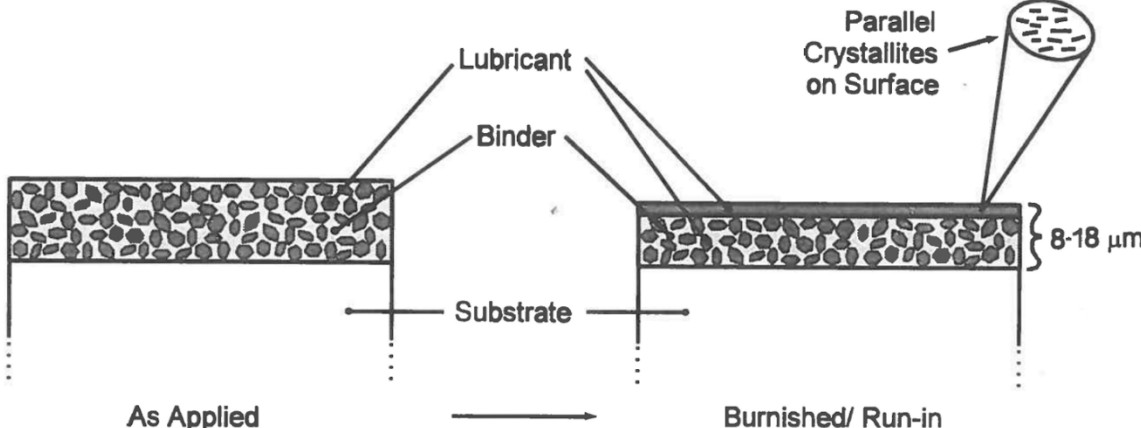

**Figure 3.** Schematic cross section of a typical bonded $MoS_2$ coating. The cross section on the left is for the as-applied coating. On the right, a cross section is shown for a coating that has experienced sliding contact. This burnishing process produces a thin, $MoS_2$-rich, crystallographically oriented film at the surface of the coating.

Appropriate surface pretreatment is critical for bonded coatings, since poor pretreatment can cause flaking and peeling of the coating even before burnishing and subsequent use. Surface preparation and cleaning procedures for bonded solid lubricant coatings are summarized in Table 7.2 in Ref. [2].

Besides adequate precleaning, surface roughening may be used to optimize adhesion, although roughness should not be so high that abrasive wear occurs. This is commonly done by grit blasting using alumina, sand, or steel grit. Optimum surface roughness is in the range 0.5 to 0.9 μm (20 to 35 μin) rms, best achieved using 120 [47,48] or 220 [42,46] mesh alumina powder. The air pressure is usually in the range 10–100 psi, depending on the substrate material. After grit blasting, care must be taken to remove any abrasive residue from the surface, since this can also cause abrasive wear. Generally ultrasonicating in an appropriate solvent is adequate to remove the excess abrasive.

After cleaning and roughening, a final treatment may be done to the part's surface, usually involving a coating (most metals) or passivation treatment (stainless steel). Coatings used vary, depending on the metal used in the part. For resin-bonded coatings, such coatings include chromates, phosphates, or anodization. Similar pretreatments can be used for inorganic- and ceramic-bonded coatings, but when they are to be used at elevated temperatures, phosphating and chromating are avoided because these precoating materials could decompose.

The type of substrate influences the allowable cure temperature and the type of surface pretreatment. The importance of appropriate substrate surface pretreatment is illustrated by the following two examples.

- Timken T54148 test rings were coated with phenolic resin-bonded $MoS_2$/graphite coatings [45]. Samples were tested on the LFW-1 test apparatus (block sliding on ring) at 72 rpm (0.87 mm/s) and 630 lb load. With no pretreatment, the coating failed on loading. Two other samples underwent vapor degreasing followed by sandblasting, but the second one was subsequently treated to an additional phosphate treatment before coating. The first failed at $2 \times 10^4$ cycles, while the phosphate-treated surface lasted to $6.7 \times 10^5$ cycles.
- Journal bearing tests were conducted using phenolic resin-bonded $MoS_2$ coatings [49]. Contact pressures were 3–4 ksi, with 0.87 mm/s sliding speed. In that study, coatings deposited on 304 CRES exhibited wear lives 3–10 times greater than those on 440 C CRES. Both steels underwent the same pretreatment (grit-blasting followed by passivation). However, the 304 CRES surface was rougher, so that the adhesion of the lubricant coating was greater.

SAE AS1701—the general standard covering bonded solid lubricant coatings (including both organic- and inorganic-bonded, as well as heat- or air-cured)—specifies that the average film thickness,

based on six readings minimum, of the cured film shall be between 0.0003 and 0.0005 inches, with no single reading less than 0.0002 inches or greater than 0.0007 inches [50]. If the coating is too thick, it is more structurally weak and can flake or peel. If the coating is too thin, wear can cause premature failure. In low load conditions (i.e., less than ~1 ksi), wear life increases with coating thickness, while for higher loads (i.e., more than ~10 ksi), wear life tends to decrease with coating thickness [51]. As such, a thicker coating of 0.5–0.7 mil (12.5–18 μm) is recommended for lower loads, while a thinner coating of ~0.3 mil (7.5 μm) is more appropriate for higher loads [46].

The coefficients of friction for bonded coatings are generally in the range 0.03–0.1 for a wide range of conditions (see Table 2). For a specific condition and Hertzian contact stress, that range may be much smaller, even for different types of coatings. For example, a comparison of performance properties for commercial bonded $MoS_2$ lubricant coatings was made using a Falex wear tester at 12–24 ksi [45,52]. Coefficients of friction for all coatings fell in the range 0.06–0.07 under these testing conditions.

Although bonded coatings are not the optimum choice for cryogenic applications, manufacturers often specify that inorganic-bonded coatings can operate (under the right conditions) down to −250 °C, and organic-bonded coatings down to −220 °C (see Table 2). However, friction will increase as the temperature is reduced below about −50 °C (see Section 6.5).

The most common bonded coatings for space applications contain $MoS_2$ along with various binders and additives. The lubricants $WS_2$ and PTFE are also used, and small amounts of Ag, In, and Pb are occasionally added. The addition of $Sb_2O_3$ is thought to improve low-load properties of coatings [19] or to help in orienting the lubricating pigment within coatings [53] or both.

Bonded coatings can be applied by spraying, dipping, or brushing. Spraying provides a more uniform coating, but requires careful process control to perform correctly. Dipping produces coatings with the least uniformity.

Bonded solid-film lubricants can be either heat-cured or air-cured, depending on the application and the thermal stability of the part material. The relative merits of heat-cured versus air-cured coatings are presented in Table 2. In general, heat-cured coatings withstand higher loads and exhibit greater endurance. They also tend to be more resistant to corrosion or attack by chemicals/solvents.

Bonded coatings are best used in applications that use sliding (as opposed to rolling) surfaces, low sliding speeds, moderate to high contact stresses, and large clearances. As such, although they are used for many applications, they perform best for lower speed mechanisms such as gears, cams, sliding bearings, shear ties, or other sliding applications in space mechanisms [49,54]. Because of the large thickness and thickness variability, they are not appropriate for dimensionally-critical applications (e.g., those with small clearances). In addition, debris can be produced during wear of bonded coatings, so some contamination- or debris-sensitive applications might perform poorly with bonded coatings.

An example of an appropriate application is the lubrication of gears for the Space Station Remote Manipulator System (SSRMS). Contact stresses are in the range of 25 to 100 ksi. Gears were tested with both organic- and inorganic-bonded $MoS_2$ coatings and a Pb-based coating [55]. The organic-bonded $MoS_2$ coating showed no apparent wear over several million cycles, outperforming the inorganic bonded $MoS_2$ coating, which showed some wear and failed after ~$4 \times 10^5$ cycles. The Pb-based coating failed after only ~$2 \times 10^4$ cycles, and considerable wear was seen on the gear teeth.

### 3.3.1. Heat-Cured Resin-Bonded Coatings

Heat-cured resin-bonded coatings are the most widely used solid lubricant formulations. The cure temperature can be critical, since it may be higher than can be tolerated by the substrate, and using an incorrect cure temperature can impair the effectiveness of the coating. Also, TMDs such as $MoS_2$ and $WS_2$ can oxidize if the coatings are cured too long, affecting friction. Curing generally requires heating to 200 °C for one hour, although lower temperatures and longer cure times can be used for heat-sensitive substrates. Specific cure conditions appropriate for different lubricants and substrates may be obtained from the coating manufacturer; general guidelines are presented in Table 7.2 in Ref. [2].

Besides cure temperature, different binders have different tribological properties, corrosion resistance, ease of application, and operating temperatures. Phenolics and epoxies are the most commonly used binder materials. Phenolics provide good surface adhesion, and are harder than epoxies, while epoxies provide better solvent resistance. (Modified epoxy-phenolics exhibit properties of both materials.) Other materials, including silicones and polyimides may be used. Silicones can handle higher operating temperatures than phenolics, but are softer and offer only fair adhesion. Polyimides are good for higher load applications.

### 3.3.2. Air-Cured Resin-Bonded Coatings

Air-cured organic-bonded coatings provide a happy medium in performance between unbonded/impinged coatings and heat-cured bonded coatings [56]. Binders used are generally thermoplastic resins, such as celluloses, acrylics, alkyds, epoxies, vinyls, and acetates. Inorganic binders like $TiO_2$ may also be used. $MoS_2$ and to a lesser extent PTFE are commonly used in air-cured coatings as lubricants for space applications. Heat-cured coatings provide longer cycle life and are more corrosion-resistant than air-cured coatings. The main advantage of air-cured coatings is that they can be applied on a part after it is installed in a subsystem. This is useful for in situ repairs.

An important limitation of air-cured coatings is that they do not generally meet outgassing requirements for space, as detailed in the ASTM test method E595 [57]. Since this test requires heating the sample to 125 °C, air-cured coatings usually cannot be used on space hardware without a waiver. In fact, many heat-cured coatings can be cured by heating to a temperature only slightly higher than for the test (i.e., 150 °C), and they are preferred for space applications. In general, heat-sensitive materials used in space hardware may require a lubrication formulation other than bonded coatings.

### 3.3.3. Inorganic-Bonded Coatings (Nonceramic)

The main inorganic binders used for space applications are silicates (e.g., $Na_2SiO_3$) and phosphates (e.g., $AlPO_4$), although aluminates, organometallics, and other compounds have been used. There are several advantages of inorganics over organics as binders. For example, they are useful for applications requiring liquid oxygen compatibility [41]. In addition, they can also tolerate moderately elevated temperatures (inorganics generally can be used at temperatures up to 650–750 °C, depending on the formulation [19]), which is unimportant for space applications except some propulsion systems (e.g., low cycle bearings on launch vehicles). In addition, because they are harder than organics, inorganic binders can tolerate greater loads (i.e., $S_m$ ~150 ksi vs. ~100 ksi for organics, as per Table 2) [1]. However, because they are more brittle and wear more easily, inorganics may exhibit lower cycle life than organic binders. Examples include various types of gears and low cycle bearings [41]. A significant limitation is their tendency to soften in the presence of water/humidity (discussed in Section 6.1.3). The main lubricant mixture used with inorganic-bonded coatings is, again, $MoS_2$. Other materials such as PbS and various metals are sometimes added.

An example of where inorganic bonded coatings are used is the Mars Science Laboratory (MSL) Curiosity Rover Main Differential Pivot. In a trade study, a number of bushing materials and inorganic-bonded $MoS_2$ coatings were evaluated. A coating with a phosphate binder was chosen because it delivered the lowest frictional torque during testing [58].

### 3.3.4. Ceramic-Bonded Coatings

Ceramics are used as binders for high-temperature applications (i.e., up to 1100 °C), where resin-bonded and even inorganic-bonded formulations will not work. Since such temperatures are higher than those usually required for space-based applications, these coatings will be discussed only briefly here. They are discussed in more detail in Refs. [45,59].

TMDs such as $MoS_2$ cannot withstand such high temperatures, so other lubricants are used, including Group IIA fluorides such as $CaF_2$, $MgF_2$, or $CaF_2/BaF_2$ eutectic mixtures. PbO can also be used. Aluminum phosphate is often added for strength. Although there are exceptions, resin-bonded

coatings outperform ceramic-bonded coatings at lower temperatures. Also, most ceramic-bonded coatings require curing at temperatures above 500 °C [19], so only refractory metals and alloys, or other high-temperature materials can be used.

The PS series of high temperature solid lubricant coatings has been developed over the last 40 years by NASA, utilizing the plasma spray technique to deposit the coatings [60]. They are based on nickel superalloy binders, glass or ceramic hardeners, and a mixture of silver and alkaline metal fluorides as the solid lubricants. Over wide temperature ranges, thermal expansion mismatch between coating and substrate has proved challenging especially for microturbine engine air foil bearings. The latest version — PS400 —addresses this issue, and has been proven for temperatures above 760 °C [61]. It is comprised of NiMoAl, $Cr_2O_3$, silver, and $CaF_2/BaF_2$ eutectic.

### 3.4. Sputter-Deposited Coatings

Sputter-deposition uses a plasma sustained between a substrate to be coated and a target comprising the coating material. This is accomplished inside a pumped vacuum chamber to achieve low pressures necessary for the plasma formation but also to minimize contamination from atmospheric impurities. Both DC and RF plasmas are used. The plasmas are controlled to allow a net transport of material from the target to the substrate. This net transport is controlled by varying the polarity and magnitude of the DC voltage, or by using diode-type arrangements to rectify the RF field. Some substrate material is "back-sputtered" during deposition, which results in some intermixing at the interface between the substrate and the coating. As such, the coatings are highly adherent, which serves to increase wear life (see Figure 4). RF sputtering is more versatile in that coatings can be deposited on electrically insulating materials. By using magnetron sources, increased deposition rates and confinement of the depositing material can be achieved. The magnets can be arranged to give varying currents of ionized species onto the growing film, which can improve coating adhesion, microstructure, and density. Newer techniques like HIPIMS use pulsed voltages that also can improve microstructure/density.

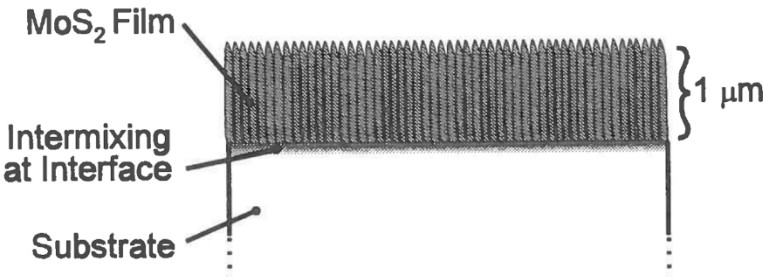

**Figure 4.** Schematic cross section of a sputter-deposited $MoS_2$ coating. This image represents a pure $MoS_2$ coating with perpendicular-oriented crystallites (see also Figure 5a). Many modern coating formulations are sputter-deposited along with other species to produce an amorphous nanocomposite material with improved properties; such nanocomposites show less discernible structure (see Figure 5c,d).

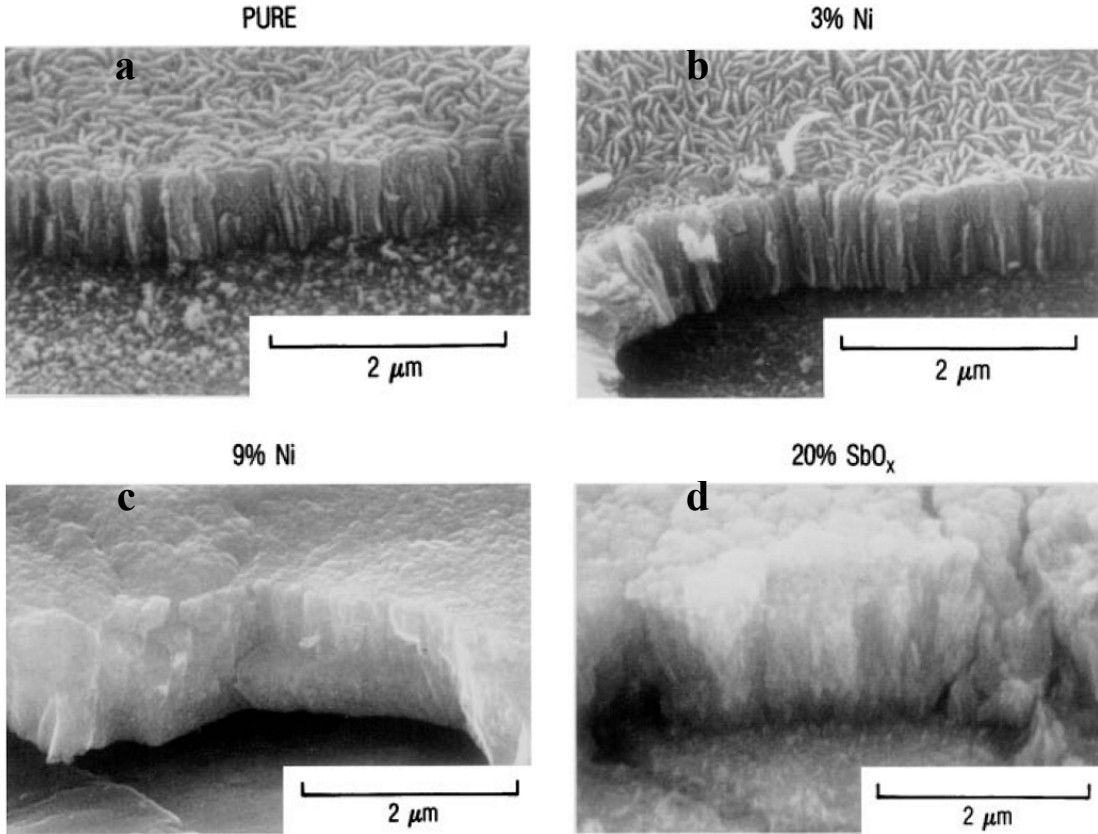

**Figure 5.** Cross sections of sputter-deposited MoS$_2$ coatings produced by indentation using a Brale indenter. Shown are (**a**) a pure coating, (**b**) a coating co-sputtered with 3 at% Ni, (**c**) a coating co-sputtered with 9 at% Ni, and (**d**) a coating co-sputtered with 20 at% Sb$_2$O$_3$. Reprinted from Hilton, M.R.; Jayaram, G.; Marks, L.D. *J. Mater. Res.*, **1998**, *13(4)*, 1022; ©Cambridge University Press, reproduced with permission.

Originally, sputtered MoS$_2$ coatings were applied using an RF technique, without the addition of other species to the coating [62]. Refinements to the sputtering technique in recent years has resulted in improvements in sputtered MoS$_2$-based coatings for many applications. In particular, the closed field unbalanced magnetron sputter ion plating (CFUBMSIP) technique was developed by Teer Coatings Ltd. to produce metal-MoS$_2$ coatings, especially Ti-MoS$_2$ coatings known under the trade name MoST$^{TM}$ [63]. The term "unbalanced" indicates an arrangement of the source magnets to produce a significant ion current on the growing coating. The ion bombardment results in better adhesion, as well as denser coatings that exhibit higher hardness and load bearing characteristics. They are much less sensitive to atmospheric water vapor during tribological testing at high humidity than other coatings. The coatings have been used successfully for a wide range of terrestrial uses like cutting and forming applications. Other metal additions such as Cr, W, Mo and Zr have been studied with varying compositions and substrates produce similar results. However, as of this writing, there is no documented demonstration of CFUBMSIP metal-MoS$_2$ coatings for spacecraft use.

There are far fewer suppliers of sputter-deposited solid lubricant coatings used for spacecraft than for the more traditional bonded coatings. There is a reticence of many manufacturers to include sputtered coatings more widely, partly due to a mass of confusing information in the literature. Although there are a myriad of literature sources on the properties and performance of sputtered MoS$_2$-based coatings, many concentrate on new formulations (i.e., cosputtered with different compounds and metals) and relatively few concern the formulations that have demonstrated space qualification/use. In addition, the deposition of sputter-deposited coatings requires high vacuum plasma deposition equipment, while bonded coatings generally require only an air brush and oven.

The properties of sputter-deposited $MoS_2$-based coatings are critically dependent on adhesion, microstructure, and composition [64,65]. To optimize adhesion, fine abrasive polishing may be performed to produce a smooth surface, followed by ultrasonic degreasing in appropriate solvents. In fact, there is evidence that light roughening of substrate surfaces prior to coating can increase endurance of sputter-deposited $MoS_2$ coatings during operation [66]. For 52100 steel coated with 1-μm-thick $MoS_2$ coatings, a plot of endurance versus surface roughness produced a curve with a maximum at $R_a$ ~0.2 μm (8.0 μin). (This optimum roughness is 3–5 times lower than that for bonded solid lubricant coatings; see Section 3.3.)

Once the substrate is installed in the vacuum chamber, oxides and other contaminants may be removed from its surface by inert gas ion bombardment using an ion gun, or by adjusting the plasma so that there is a net removal of material from the substrate rather than from the sputtering target. The etching removes the oxide layer on the surface to expose the more chemically-reactive metallic species underneath [64], which also enhances adhesion.

Coating density is an important property, since more dense coating morphologies result in decreased friction, increased endurance, and reduced environmental sensitivity. However, without the addition of other chemical species during coating growth, the resultant smaller crystallite sizes can result in poorer oxidation resistance related to extensive terrestrial storage. Dense coatings with small crystallite sizes are favored by factors that inhibit grain growth, including low substrate temperature, dopants that inhibit adsorbed atom mobility, lower growth pressures, and/or ion bombardment during growth (e.g., Ion Beam Assisted Deposition and CFUBMSIP).

The composition of the coating is the third critical parameter and encompasses both the intentional and unintentional addition of species. Unintentional introduction of additional species in sputter-deposited $MoS_2$ coatings usually takes the form of oxygen incorporated during sputter-deposition and after exposure to atmosphere. Because the Mo-O bond is stronger than the Mo-S bond, the presence of relatively small amounts of water vapor in the chamber during deposition (as low as $10^{-6}$–$10^{-5}$ Torr) can result in appreciable amounts of oxygen being incorporated in the coatings [67–70]. Coatings may contain as much as 15% oxygen. This is especially true when coatings are deposited in a chamber that must be opened every time new samples are loaded; adsorbed water is readsorbed on the surfaces of the chamber and sputtering source each time they are exposed to atmosphere. If the water vapor pressure is not too high, a mixture of poorly crystalline $MoS_2$ and $MoS_{2-x}O_x$ phases results in the coatings. The $MoS_{2-x}O_x$ phase (where $x$ is continuously variable) has an $MoS_2$-like structure, with oxygen atoms substituted for sulfur atoms in the $MoS_2$ crystal lattice. (Because of the sulfur depletion, some previous studies have assumed that coatings were present in a $MoS_x$-type composition, but these studies did not adequately account for the presence of oxygen in the coatings.) Most sputter-deposited coatings in production today contain appreciable amounts of this phase. Low-oxygen coatings can only be produced by using a sample introduction chamber so that the deposition chamber remains under vacuum at all times to preclude water adsorption into the $MoS_2$ target. Even if they are deposited in a highly pure sputtering ambient, if the coatings are substoichiometric (i.e., $MoS_x$) the coatings can absorb some oxygen-containing species to produce the $MoS_{2-x}O_x$ phase after brief exposure to atmosphere [22]. The presence of this oxygen-containing phase is not necessarily detrimental, but should be understood and controlled during coating growth since varying oxygen contents can affect the microstructure and friction coefficient in the coatings. In fact, friction may decrease with small values of $x$ in $MoS_{2-x}O_x$), [68,69,71] although endurance may not. However, a sufficiently high water component in the sputtering ambient could lead to $MoO_3$ formation, which would be detrimental to both friction and wear life. The detrimental formation of $MoO_3$ can also occur during prelaunch storage; this is discussed in Sections 6.1.1 and 6.1.2.

As discussed above, improvements in the tribological properties of sputter-deposited $MoS_2$ coatings have been achieved using intentional doping by cosputtering the $MoS_2$ with other species to form a nanocomposite [72,73]. Two or more materials can be deposited simultaneously from separate targets or deposited from a single target comprised of a mixture. However, depositing from a single

mixed target can give a coating composition different from the target composition due to preferential sputtering, especially during initial sputtering with a new target.

Such nanocomposite coatings have been shown to exhibit desirable tribological characteristics, including enhanced wear life as well as lower and more stable friction [74–78]. Metals and other species can densify the coatings by poisoning the edge of the $MoS_2$ crystallites during growth of the coatings [64,79,80] (compare Figure 5a,c,d). In addition, the nanostructure results in a harder, more fracture tough coating that resists wear [69,78]. Added species can also reduce the oxidation of the $MoS_2$ crystallites themselves by acting as sacrificial oxidants, and by "sealing" the reactive edges of the crystallites [79]. In illustration, $Sb_2O_3$-Au-$MoS_2$ coatings have been shown to exhibit minimal wear, even when tested in 50% RH air; in contrast, pure $MoS_2$ coatings exhibited rapid wear under the same conditions [81].

Nanostructure aside, it seems counter-intuitive that mixing a relatively poorly lubricating material with a solid lubricant will improve performance. However, sliding contact produces a thin surface layer on the surface of the nanocomposite coating whose composition is virtually pure $MoS_2$. As for bonded $MoS_2$ coatings (see Section 3.3)—and in fact for all $MoS_2$-based composite materials—the surface layer consists of $MoS_2$ crystallites with their basal or (0001) crystallographic planes aligned parallel to the coating surface. The result is a highly lubricious surface layer that sits on top of a harder, fracture tough coating that can withstand high loads.

This process was demonstrated by conductive atomic force microscopy (c-AFM) of a sputtered Au-$MoS_2$ nanocomposite coating (with 75 at% Au) [82]. Images were obtained after varying numbers of cycles/frames from 1 to 30 (see Figure 6a). The first frame shows a mixture of nanosize Au and $MoS_2$ domains at the surface. As the AFM tip was scanned over the surface, the amount of visible Au particles decreased rapidly, so that after 30 frames, they were virtually nonexistent, and the surface was covered mostly with $MoS_2$. Figure 6b represents a topographic AFM image taken after 30 cycles that shows that the initially amorphous $MoS_2$ phase has crystallized due to the sliding interaction.

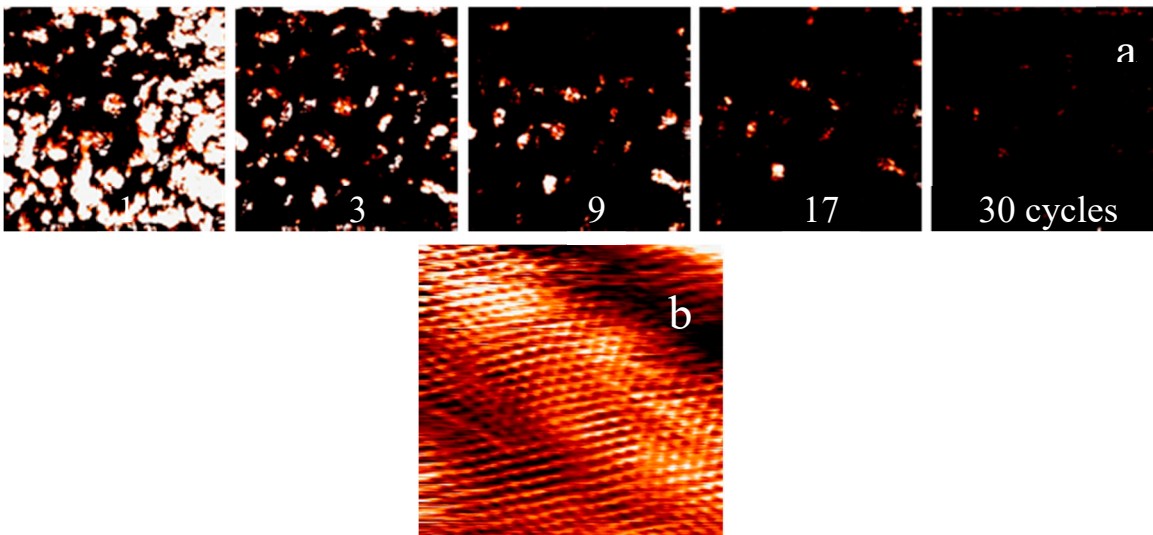

**Figure 6.** (**a**) A series of AFM conductivity images measured over a 100 nm × 100 nm area. Five image frames (1, 3, 9, 17, 30) were selected from a series of 30. The initial frame (**far left**) consists of relatively evenly dispersed domains of Au and $MoS_2$, but in later frames the Au progressively disappears and the $MoS_2$ domains appear to progressively spread across the surface. Near complete surface coverage by $MoS_2$ is suggested by the final frame (**far right**). (**b**) A topographic image (7 nm × 7 nm) of a sliding-induced $MoS_2$ tribofilm showing hexagonal structure and a lattice spacing of approximately 0.31 nm. This lattice structure is virtually identical to that of a single crystal $MoS_2$ with its basal plane parallel to the surface. Reprinted from Kim, H.I., Lince, J.R. *Tribol Lett,* **2007**, *26*, 61–65, with permission of Springer Nature.

POD wear testing of a Au-MoS$_2$ coating (with 59 at% Au) shows similar evolution of the surface region. After testing (and before coating failure), Auger Electron Spectroscopy (AES) analysis sensitive to the topmost 1–2 nm of the surface was conducted in the wear track. The results showed enhancement of S from the MoS$_2$ on the surface of the wear track, with virtual disappearance of the Au [30]. Other studies have demonstrated this effect with composite PVD MoS$_2$-based coatings of varying compositions. For example, Micro-Raman spectroscopy demonstrated the formation of crystalline MoS$_2$ on the surface of amorphous IBAD Pb-Mo-S coatings [83].

The main nanocomposite MoS$_2$ formulations that have demonstrated space heritage are Ni-MoS$_2$, Sb$_2$O$_3$-MoS$_2$, and Sb$_2$O$_3$-Au-MoS$_2$, although pure MoS$_2$ coatings continue to be used. (Pure coatings may exhibit lower friction in very early life compared to nanocomposites, at the expense of lower cycle life).

Sputter-deposited nanocomposite Sb$_2$O$_3$-Au-MoS$_2$, Sb$_2$O$_3$-MoS$_2$, and Ni-MoS$_2$ coatings were tested in a POD friction/wear study conducted in nitrogen (<0.08% RH) [12]. The Sb$_2$O$_3$-Au-MoS$_2$ and Sb$_2$O$_3$-MoS$_2$ coatings were shown to greatly outperform the Ni-MoS$_2$ coating, by an average of 18× and 26×, respectively. In addition, the steady state friction for Ni-MoS$_2$ was slightly higher than for the Sb$_2$O$_3$-containing coatings. The difference in tribological performance may be explained partly by the significantly higher amount of cosputterant in the Sb$_2$O$_3$-containing coatings relative to the Ni-MoS$_2$ coating, since a major effect of forming the nanocomposite is to densify and reduce crystallinity. As discussed above, this results in a harder, more fracture tough coating that resists wear. Simply raising the Ni content to similar values of Sb$_2$O$_3$ should not result in improved tribology: Sb$_2$O$_3$ as an additive is likely successful at higher concentrations because it is relatively soft (<100 MPa Vickers Hardness for Sb$_2$O$_3$ [84] vs. >600 MPa for Ni and other transition metals), as well as being easier to burnish into thin films [78].

It was once thought that sputter-deposited MoS$_2$ coatings would always be the "lubricant of the future," but they are being used increasingly to lubricate components in space programs, including ESA's Infrared Space Observatory (ISO) [3,85], ENVISAT-1 [86], TRIAD [41], JWST [13,14], NASA ISS [87], MSL sample collection drill [88], Rosetta [89], BepiColombo (MPO) [90] and many commercial and US government space vehicles. The coefficient of friction of sputter-deposited MoS$_2$ coatings is considerably lower than that for most other solid lubricant formulations; the resultant decrease in torque of the lubricated device is attractive considering the need to reduce power budgets on spacecraft. Also, sputter-deposited coatings are good choices for dimension-critical applications, since their thickness and thickness variability are small relative to other formulations (see Table 2). Although pure sputter-deposited MoS$_2$ coatings may exhibit more moisture-sensitivity than resin-bonded coatings, the newer nanocomposites containing added metals and other compounds exhibit greater robustness with respect to humidity, during storage and testing (see Section 6.1.2). A main drawback to usage of these coatings is the difficulty in coating large parts. Some also see the cost as prohibitive, although a few thousand extra dollars is not excessive if it provides increased confidence for a mission that costs hundreds of millions of dollars.

### 3.5. Metal Coatings

Metal coatings, primarily those made of lead (Pb), reduce friction in ball bearings by providing low-shear flow in the contact region during microslip. Lead's low-shear properties arise from a high density of dislocations within its FCC structure [11]. Its shear strength is higher than that for MoS$_2$ and so generally gives higher friction in vacuum. One advantage of lead coatings over those of silver or indium is the unavoidable presence of PbO, a reputed good solid lubricant, within the coatings (see Figure 7). However, the upper temperature limit of using lead is limited because of its low melting point (see Table 2).

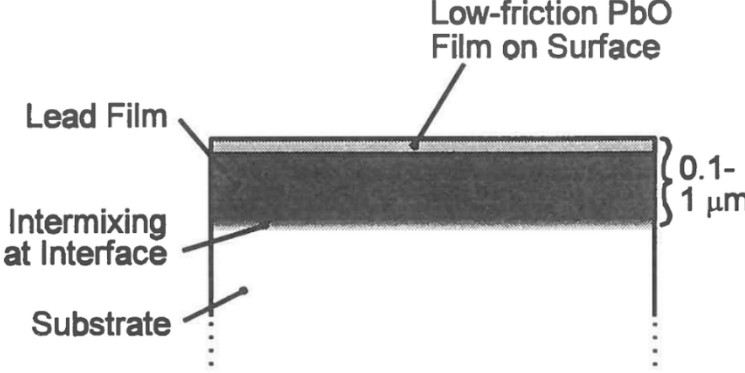

**Figure 7.** Schematic cross section of an ion-plated or sputter-deposited lead (Pb) coating.

Metal film lubrication must be applied by a process that ensures adequate adhesion to the substrate surfaces, to avoid immediate loss from the contact. Coatings can be deposited by electroplating, evaporation, sputter-deposition, and ion-plating. Ion plating is likely the most common method for applying metal films and lead in particular [16], but sputter deposition has also been used successfully [91]. Sputter-deposition and ion-plating give coatings with good adhesion, while allowing control over coating thickness and uniformity. In addition, compositions and morphologies are more controlled than they are for the electroplating process.

In ion-plating, ionized metal atoms are produced by evaporating them into an inert gas plasma. The part to be coated is biased negatively, attracting the positively charged metal ions to its surface to form the coating [92]. The structural and morphological properties of the coatings can be tailored for conditions of use by changing the bias and power settings.

Surface preparation for ion-plated samples is similar to that for sputter-deposition. The sample is finely polished to produce a smooth surface and degreased. Also similar to sputter-deposition, the sample can be ion-cleaned in the gas plasma before coating deposition to enhance the chemical bonding between the coating and substrate.

Thin metal coating lubrication is useful mainly for rolling rather than sliding applications [46,92,93]. Coating thickness typical falls in the range 0.1 to 1 μm, which corresponds to both maximum wear life and minimum coefficient of friction [46,92]. Ion-plated Pb coatings have proven especially useful for rolling element bearings involving relatively slow rates of rotation in space mechanisms (e.g., solar array drives) [92,93]. Such Pb coatings have been shown to exhibit significantly greater cycle life than sputtered $MoS_2$ coatings in ball bearing applications, although their steady state friction is 2.5× greater as measured in the spiral orbit tribometer (SOT) [94].

Besides Pb, the noble metals Ag and Au have been used. For example, an Au(Co) alloy coating was used for lubricating ring tracks and rolling flexures in the roll ring assembly on Space Station Freedom [95]. Also, Ag coatings have been explored as a solid lubricant for ceramic coatings (i.e., $Si_3N_4$ and $ZrO_2$) [96]. When tested in pin-on-flat contact with a light mineral oil at ~150 ksi Hertzian stress, the Ag coatings exhibited coefficients of friction 2–3 times lower than for uncoated ceramic surfaces, and reduced wear rates to negligible levels.

### 3.6. PVD/CVD-Deposited Hard Coatings

Although not solid lubricants *per se*, hard coatings including metal carbides and nitrides deposited using PVD and chemical vapor deposition (CVD) techniques can provide significant wear reduction, while also exhibiting lower friction than uncoated part surfaces. These materials are not commonly used in space [97], partly because the bulk materials are brittle. However, when applied as thin coatings where the materials properties of the substrate dominate, they are useful for niche applications.

In space, titanium-based hard coatings have been used more often than other hard coating materials. TiC and TiN are useful because their COF values are not greater in the vacuum of space

compared to air. For example, TiC and TiN show COF values (against steel) that are as low or lower in vacuum (0.15 and 0.27, respectively) than in air (0.26 and 0.29) [97]. In addition, these values are considerably lower than metal-on-metal unlubricated friction, where COF values often exceed 1.0. TiC also has the advantage of having higher hardness and elastic modulus than other carbides, nitrides, and borides [98]. Finally, titanium-based hard coatings are particularly attractive for coating titanium alloys like Ti6Al4V, which is used extensively on spacecraft because of its strength and it has lower mass density than steel: the common element titanium in both coating and substrate ensures a strong adhesive bond.

Balls in spacecraft bearings have been coated with TiC via chemical vapor deposition (CVD), which has been useful for mechanisms including deployment and release mechanisms, and gyroscopes [99]. They may be especially useful in hybrid bearings with increased temperature variation during operation: balls made from silicon nitride are typically used in hybrid ceramic-steel ball bearings, but the CTE mismatch between TiC and steel is smaller than that between $Si_3N_4$ and steel [100]. A proviso of the TiC CVD coating process is the balls must be held at temperatures ~1000 °C during deposition. As a consequence, stainless steel balls must be rehardened and polished after coating by this process to restore sphericity.

TiCN has been used in space mechanisms because it exhibits both high hardness, as well as lower COF than TiN and other similar hard coatings [101]. The wear of disks lubricated with Rheolube 2000 grease during POD testing in a dry $N_2$ environment was significantly lower for TiCN-coated disks than for uncoated disks [102]. Another POD study conducted in ultra-high vacuum without lubrication showed that TiCN had lower friction than TiN (0.45 vs. 0.62) [103]. TiCN has demonstrated space heritage in several applications for use as a coating on titanium alloys as a counterface to solid-lubricated steel surfaces, as discussed in Section 4.

There are, of course, a myriad of metal carbides and nitrides to choose from for spacecraft mechanism surfaces. These are used in terrestrial environments for applications including cutting and forming tools. There have been extensive tribological studies, but very few were conducted in vacuum or inert environments [104], which is a requirement for determining effectiveness when used unlubricated in space applications. However, other attributes may point the way to coating materials that can be chosen for further study in a spacecraft environment.

For example, rolling contact fatigue (RCF) life is an important variable to consider for long duration, high speed gyroscope ball bearings. In a review of RCF life of a number of different coatings, it was found that coatings produced using different techniques exhibited RCF life in the following order: PVD > CVD > Thermal spray [105]. It was found that compressive residual stress contributed to increased fatigue life in PVD coatings, but only for coating thicknesses ≤1 μm. HfN was a standout in one study. But in many cases, identifying the best specific compounds was more difficult. For example, CrN, and Ti(CN) performed well under many conditions. However, sometimes Ti(CN) had a higher RCF life than CrN, and sometimes CrN was higher. The results for these and other coatings in the review clearly depend on coating deposition conditions, coating thickness, $S_m$/load during testing, and the presence/absence and type of lubricant present. Inert and vacuum conditions were not used in the studies.

### 3.7. Ion-Implantation

The surfaces of steel or other bearing materials can be bombarded with medium energy ions (i.e., ~50–200 keV ion energy), primarily to enhance corrosion resistance [106]. However, tribological performance is also enhanced: wear is lowered by increasing surface hardness, but friction may also be lowered by producing lower friction species at the surface. The ions are implanted within a range of approximately 0–200 nm (0–8 μin) into the surface by an ion accelerator (see Figure 8). Within that depth, the atomic concentration of the ions can be as much as 10–50 at%. Ions can also be implanted via plasma immersion ion implantation (PIII) [107]. A wide variation of materials can be implanted, including metal ions such as Mo, Sn, Pb, and In, as well as ions such as B, N, P, S, and C,

and even inert gas ions such as Ar and Kr. The degree of tribological improvement is dependent on the substrate material, implanted ion, and the ion energy and fluence. Fluences of $10^{16}$–$10^{17}$ ions/cm$^2$ are usually optimum.

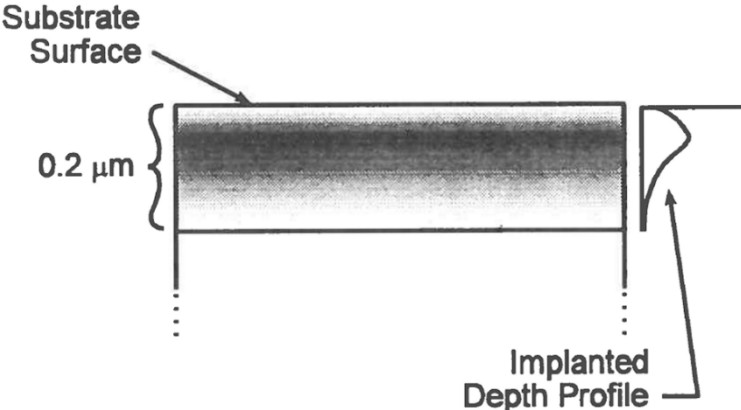

**Figure 8.** Schematic cross section of the surface of a metal that has been implanted with ions to produce improved corrosion-resistant and tribological properties, including hardness.

Plasma nitriding is often used in space applications, particularly for gears [11]. It modifies the mechanical properties of the material to yield a harder, wear resistant surface that also deforms less under load. Reduced deformation allows any subsequently-applied solid lubricant coatings to experience less strain, improving endurance. Increased hardness is likely due to the formation of hard metal nitrides [108]. Another possible reason is the presence of an amorphous layer produced by implantation; such a layer would inhibit crack propagation, increasing fatigue strength.

### 3.8. Composite Materials

Self-lubricating materials are used in space mechanisms primarily as cage materials in ball bearings and as bushings in journal bearings. They generally contain the lubricants PTFE and/or MoS$_2$, and structural materials such as pure polymers (e.g., polyimides [109]), composites (e.g., PTFE with glass fiber reinforcement [19]), or metals [110]. [In low contact stress (S$_m$) applications, PTFE can act as a self-lubricating material without the need for reinforcement.] The best structural materials for minimizing wear are strong, but softer than the bearing ball and race material.

A commonly-used bearing cage composite for space applications is composed of PTFE and MoS$_2$ lubricant powders contained in a glass fiber matrix for reinforcement. It is generally accepted that the PTFE forms the transfer film on the ball and race surfaces. The addition of the MoS$_2$ serves to minimize wear of the balls due to contact with the glass fibers [111].

Metals are also used in composite bearing cages for space applications; they provide reinforcement for the lubricant filler, but are usually softer than the steel bearings in which they are used. A common example is bronze, which is generally used with 20–60% PTFE. MoS$_2$ may be used in addition to PTFE, but usually in smaller amounts (i.e., ~5%).

In manufacturing metal/solid lubricant composites, several technologies can be used, but PTFE- and MoS$_2$-containing composites are generally formed using powder metallurgy [110]. Powder metallurgy involves mixing the various components, followed by compacting at high temperatures, and then sintering. Larger and more spherical particles optimize mixing and avoid segregation, as does the use of materials that have similar densities.

Although not composites, some monolithic polymer materials are included in this category because they are also used for bearing cages and bushings. These polymers are designed to provide a balance between strength and low friction, and include polyimide (PI), poly(amide-imide) (PAI), and poly(aryletherketone) (PEEK), as well as pure PTFE. In bushings and sleeve/journal bearings, the contact is over a large surface area and contact stress is relatively low. In such direct contact

applications, some transfer of lubricating material may take place, but lubrication is accomplished mainly by providing a low friction interface between the (composite or polymer) sleeve and the (steel or other metal) rod.

An example of an appropriate application for solid PTFE is in the boom deployment mechanism and solar sail spool assembly for the Near Earth Asteroid (NEA) Scout mission [112]. They were used as bushings on rotating shafts instead of ball bearings to save space, and worked because of the limited number of cycles and relatively low values of the maximum Hertzian contact stress ($S_{max}$).

Composite bearing cages provide low friction at the ball/cage interface, where contact stress is low. In addition, they provide low friction/wear at the high contact stress ball/race interface by lubricant transfer: first from the cage to the ball, and then from the ball to the race [113] (see Figure 9). Transfer is inefficient, so a high percentage (>20%) of lubricant in the composites is necessary. The result is poor mechanical strength and higher wear rates. In some cases large particles (lumps) of material can be transferred causing noisy operation.

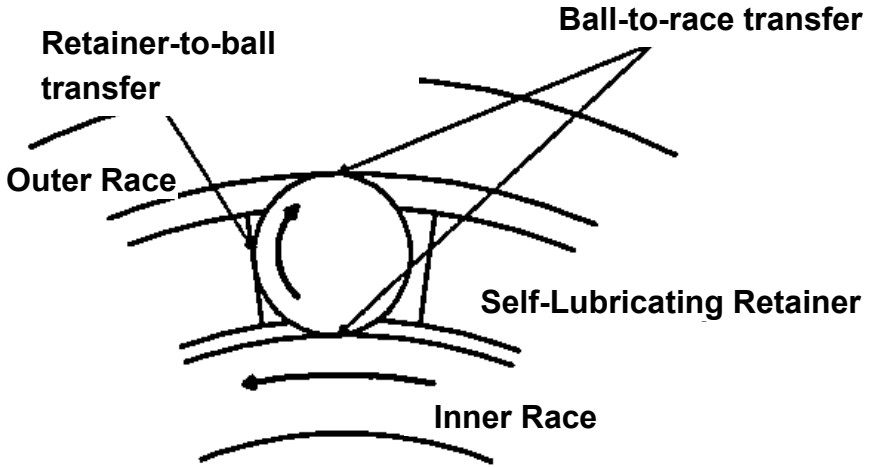

**Figure 9.** Cross-sectional schematic representation of a self-lubricating bearing. Lubrication occurs via transfer of lubricant, first from the cage to the balls, and then from the balls to the races. The cage may be composed of a monolithic lubricating material, such as polyimide or PTFE, or more commonly of a composite mixture of one or more lubricants (e.g., $MoS_2$ and/or PTFE) with a stronger material for structural integrity (e.g., glass fibers, bronze, or polyimide).

For polymers, wear rates in the early stages of sliding vary with roughness as $(R_a)^n$, where $n$ = 2–4 [46]. Therefore, smooth counterfaces are recommended for dry bearings. However, the wear rate rapidly drops as the lubricant is transferred. For polymer composites that contain solid lubricants, wear behavior is highly dependent on initial transfer films, which in turn are dependent on environmental factors. Relative humidity can either increase or decrease wear depending on the type of filler. Therefore, to avoid processing variations and performance uncertainty, bearings to be used on spacecraft should be run in (worn) under a controlled environment (preferably vacuum or dry $N_2$). To aid in design, approximation schemes exist for predicting wear rates for a particular application [46,114]. Such schemes take into account the specific wear rate of the material (i.e., in $m^2/N$), and include factors for geometry (continuous vs. oscillatory, movement of load on bearing), heat dissipation, operating temperature, counterface material, and counterface roughness.

There is considerable evidence that the use of thin lubricant coatings on the races (and sometimes balls) along with composite cages significantly improves performance and endurance as opposed to using either the composite or the coating alone. The coating allows a graceful run-in process, providing initial lubrication between races and balls while the transfer films from the cage are being developed. Examples include Pb/bronze cages with ion-plated Pb coatings deposited on the bearing races [11,115],

and PTFE-based cages (using glass or bronze as a filler) with sputter-deposited $MoS_2$-based coatings on the bearing races [11,116–118].

Many instances of the use of composite materials for bearing cages can be found in the published literature; a few examples are provided here. Bronze (Salox-M) or glass fiber-reinforced PTFE (Rulon A) with added $MoS_2$ showed the lowest wear in bearings for Pratt & Whitney's RL-10 $H_2/O_2$ engine [4].

PTFE or polyimide cages with $MoS_2$ coatings on the races or Pb bronze cages with Pb on the races were tested for oscillating (gimbal type) bearings, and different levels of performance were obtained depending on the oscillation arc [119]. Pb systems were better for small arcs, ±0.5°, while $MoS_2$ was better for ±5° and ±20° oscillations.

A detailed study of polyimide cages, including fluorinated material, showed that a formulation with 7.5 vol% $MoS_2$ and fluorinated polymer gave the best overall performance: longest life with lowest frictional noise [120].

In another study, angular contact bearings were tested in vacuum at temperatures from 300 K down to 20 K [115]. Three lubricating systems were used: (1) PTFE transfer films from a Duroid 5813 (glass-filled PTFE/$MoS_2$) cage, (2) ion-plated Pb coatings on the races with a Pb-impregnated bronze cage, and (3) sputter-deposited $MoS_2$ on the races with several cages (Duroid, $MoS_2$-coated steel, and $MoS_2$/polyimide composite). Pb-coated bearings were noisier than Duroid (with or without sputter-deposited $MoS_2$), but the Pb-coated bearings showed no torque deterioration on reducing the temperature from 300 K to 20 K, while the Duroid bearings all showed minor deterioration. All of the bearings tested at 20 K survived at least 2 million revolutions. Additional low-temperature data [85] show that using sputter-deposited $MoS_2$ as the only lubricant deposited on the races, balls, and the steel cage was the best performer of all (although only mean torque values were shown; no noise levels).

An example of the excellent performance that can be attained using a combination of thin lubricant coatings with composite cages is presented in Ref. [121]. Angular contact ball bearings were lubricated with a sputtered $MoS_2$ coating on the races and balls, and used with a PTFE composite cage. The bearings were tested at a high load (i.e., $S_m$ = 222 ksi) in vacuum, and showed an endurance of $10^7$ revs. A similarly tested bearing with the PTFE composite cage—but without the sputtered $MoS_2$ coating—showed significantly higher torque noise, and lower endurance.

Duroid 5813, a glass-filled PTFE/$MoS_2$ product made by the Rogers Corp. was often used prior to the late 1990's, but because it stopped being manufactured at that time, the PGM-HT material (made by JPM Mississippi) was investigated as a substitute. Its composition is similar to Duroid, and it was found to perform similarly to Duroid in testing by ESA [122]. It is now being used extensively [10,111,123]. Additional studies by the ESA have identified composites of similar composition that could act as alternatives to PGM-HT [111,124].

There is an inverse relationship between the $S_{max}$ on the bearings and their endurance. For one bearing design lubricated with PGM-HT, Ref. [10] showed that doubling $S_{max}$ resulted in a drop in the cycle life of the bearings by a factor of ~5. This held true up to ~1200 MPa, where the cage failed rapidly because the PTFE transfer film was worn rapidly.

Thin metal coatings can also be applied as composites. For example, the interiors of telescopic tubes used in a deployable/retractable telescoping boom were coated with a PTFE-impregnated electroless Ni plating [125]. The Ni/PTFE coating lowered friction and prevented galling between the Al in the tubes and the tips of latch pins during deployment.

Another similar process is to impregnate PTFE into porous surfaces such as anodized aluminum coatings. Examples are Hardtuf (Tiodize Inc.) and Tufram (General Magnaplate). (Similar coatings can be effected on titanium alloy surfaces under the names tradenames Tiodize and Canadize [General Magnaplate].) The low friction of PTFE and the small thickness of these films is attractive, but care must be taken to use them only in low contact stress applications. Although such films are thought of as composite films, the porosity of the oxide is sometimes difficult to control, so the PTFE may be present only on the surface of the anodize layer [126]. In such a case, the process can only be used for

contact stresses 1–10 ksi, since above this level the PTFE will cold flow, followed by abrasive wear of the anodize and potential galling of the underlying aluminum.

Generally, the anodized surface should be not be sealed prior to PTFE application (sealing is accomplished by immersing the coated part in DI water at ~100 °C to fill the surface pores with hydrated aluminum oxide). The PTFE (or TFE telomer) is dispersed in a volatile solvent and applied to the surface. If the substrate is not heat-sensitive, it can be heated to the PTFE melting temperature. Melting the PTFE enhances its adhesion to the anodize coating, and therefore its endurance, although it does not increase the coating's load-carrying capacity.

Bonded $MoS_2$ coatings can also be formed on the surface of anodized aluminum and titanium surfaces. This results in low friction and potentially higher cycle life than PTFE impregnation, especially for higher $S_{max}$ values.

## 4. Optimized Solid-Lubricated Contact Design

Good design practice requires taking into account not just the type of lubricant, but also:

- Materials and chemical properties of the device surfaces
- Hertzian contact stress ($S_{max}$)
- Type and duration of relative contact motion
- Environment (gaseous/vacuum, temperature, launch)

It is especially important that the materials properties and chemistry of both contacting surfaces—as well as their interaction together—are considered when designing a solid-lubricated system. The key attribute is robustness. Solid lubricant coatings can fail—either completely or in small areas—and when this happens the system must be tolerant to metal-to-metal contact. The following guidelines can aid in ensuring this robustness.

Ideally, the two base materials should be chemically dissimilar. Should the lubricant coating fail, adhesive wear and potential galling due to metal-to-metal contact can be delayed or precluded. Rabinowicz developed a compatibility chart for a number of potential metallic element couples, summarizing the solubility of one element in another. The notion is that minimizing mutual solubility will reduce the possibility of adhesive wear/galling [127]. Although only elements are discussed there, with the addition of a little chemical knowledge, appropriate alloys as part materials can be chosen.

The Threshold Galling Stress (TGS) can also be used to determine appropriate materials to use for the two sides of a contact; this is tabulated in several places in the literature for different materials, including Ref. [128]. The TGS is the minimum Hertzian contact stress where galling is initiated; it is generally measured using two crossed cylinders, but other geometries have also been used (including POD). The higher the TGS, the more resistant the materials couple will be to adhesive wear and galling.

Chemical dissimilarity should also be taken into account when using any surface hardening techniques including the deposition of hard metal or ceramic coatings. For example, hard chromium or electroless nickel coatings are often deposited on surfaces to provide a harder, more wear resistant material. Designers are often tempted to use the same coating on both contacting surfaces, assuming that if coating one side is good, coating the other side with the same material is even better. However, as Ref. [127] shows, identical metals are the most susceptible to adhesive wear and galling. If hard coatings are used on both contacting surfaces, the two should be chemically dissimilar.

Optimally, hard coatings should be comprised of oxides, carbides, or nitrides, rather than metallic species. These compounds are not only harder than metals, but are more chemically stable with respect to opposing metallic surfaces, and therefore are less likely to promote galling. For example, anodize coatings on aluminum and titanium alloy surfaces is encouraged. The use of PVD coatings including TiC or TiN is even better. As discussed in Section 3.6, TiCN is particularly attractive, as it exhibits both high hardness, as well as a lower COF than TiN and other similar hard coatings [101].

Finally, the choice of the solid lubricant coating should be made with respect to the specific requirements, as discussed at length in Section 3. Most important to consider is the expected load

or $S_{max}$ on the contact. For example, as Table 2 shows, PTFE has a low COF, but can only be used in low load applications. $MoS_2$ can be used at higher loads, but the maximum load depends on how it is formulated. For example, heat cured, resin bonded $MoS_2$ can be used at $S_{max}$ values up to approximately 150 ksi ($S_{max}$ is 1.5× the mean value [$S_m$], averaged over the contact area), while (appropriately formulated) sputter-deposited $MoS_2$ can be used for $S_{max}$ values up to approximately 300 ksi.

An example of a robust coating design involves a launch lock (LL) mechanism produced by Sandia National Laboratory [17]. Most of the sliding/releasing interfaces in the LL are solid lubricated. The design includes a spherical interface that holds a ball firmly in a socket during launch, and then must release reliably when deployed (see Figure 10). The device failed to release during preflight testing, due to stiction at the critical ball/socket interface. Some design and tolerance issues needed to be corrected, but there was also inadequate protection against galling at the ball/socket interface. The original design used Ti6Al4V for both the ball and socket, and both were coated with Tiodize (a titanium anodize process), however the Tiodize antiwear properties were inadequate for the required environment. After testing a number of materials combinations, the final design used AM-355 steel for the ball material, and Ti6Al4V for the socket material. A TiCN PVD coating was applied to the socket surface, and a sputter-deposited $Sb_2O_3$-Au-$MoS_2$ nanocomposite coating was applied onto the ball surface. The TiCN coating is hard, relatively low friction, and adheres well to the underlying Ti6Al4V substrate. The $Sb_2O_3$-Au-$MoS_2$ coating is low friction and has a relatively high endurance for a sputtered coating (as discussed in Section 3.4). Finally, not only are the coatings chemically dissimilar, but the substrate materials are also, ensuring a higher TGS value.

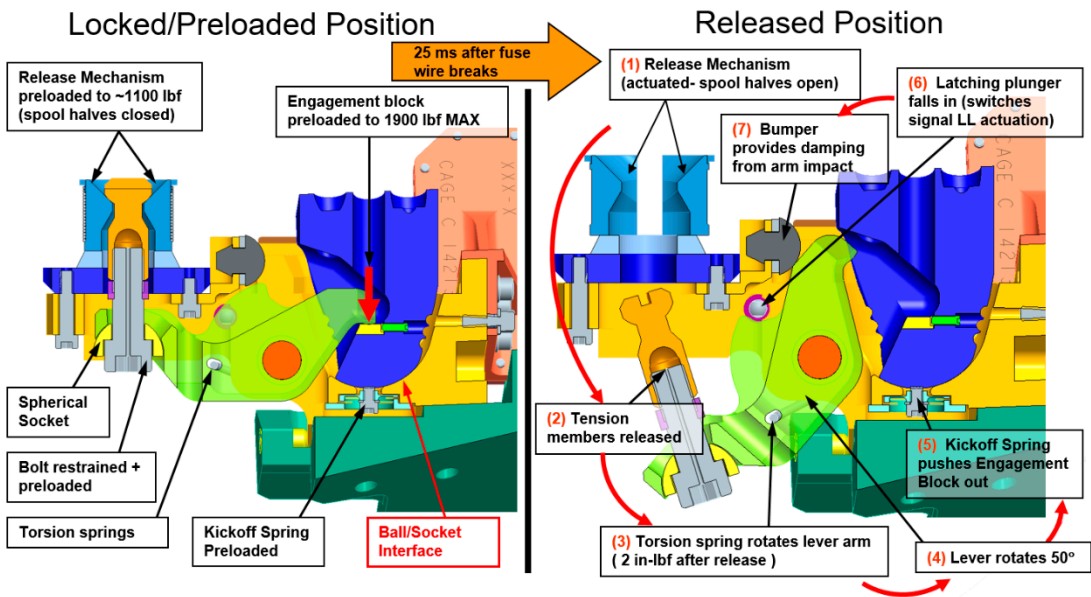

**Figure 10.** Schematic of a launch lock (LL) mechanism intended to hold a subsystem securely during launch and then release reliably on orbit. Most of the sliding/releasing interfaces in the LL are solid lubricated. The critical ball/socket interface discussed in the text is shown in red. Reprinted from Villa, D.L.; Toledo, G. Proc. 39th Aerospace Mech. Symp., Huntsville, AL, 2008, pp. 131–144, NASA-CP-2008-2152520, with permission of D. L. Villa, Sandia National Laboratory.

## 5. Testing of Lubricated Devices

There are two categories of testing pertinent to solid lubricated devices, both which are important for the spacecraft mechanism designer. The first category includes tests that are standardized for use with solid lubricants. Standard tests provide a common language for comparing lubricants produced by different manufacturers, and provide qualitative guidance in coating selection criteria for the mechanism designer. However, they do not provide quantitative friction/torque, wear rate, or

endurance life values for a specific application. As such, any lubricated device should optimally be subjected to the second category of testing, which duplicates the conditions of the actual device as closely as possible.

*5.1. Standard Tests*

Standard performance testing of solid lubricant systems is important for quality control during lubricant manufacture, for providing quantitative criteria for manufacturing specifications (e.g., AIAA, SAE, Milspec, and NASA specifications), and when investigating a trade space of lubricating formulations for a new application. Examples of standard wear tests are listed in Table 7.5 in Ref. [2], and discussed in detail in Refs. [43], [47], and [48]. More common tests include the POD (i.e., ball-on-flat) as well as block-on-ring contact geometries. Even in carefully controlled conditions, there can be significant scatter in the results. Therefore, it should be emphasized that a number of tests should be performed for a single set of variables to characterize the level of error in the results. For example, with the Timken apparatus, scatter can exceed ±100%, while with the Falex test, scatter tends to be less than ±50% [46].

These tests are typically conducted in air or dry $N_2$. Air is especially problematic—especially for $MoS_2$-containing formulations—because of the effect of humidity on performance (see Section 6.2). Dry $N_2$ is often used because it excludes most moisture and $O_2$. However, there is controversy on how well dry $N_2$ mimics the vacuum space environment, as well as on the surface chemical bases of potential differences. For example, pure sputtered $MoS_2$ and CFUBMSIP Ti-$MoS_2$ samples were tested via POD tribometry in air, $N_2$, high vacuum, and ultra-high vacuum, and differences were found between all environments. For pure $MoS_2$, high vacuum gave the lowest friction, while for Ti-$MoS_2$, $N_2$ gave the lowest friction [129]. In another study, the friction of a sputtered $MoS_2$ coating was found to be similar between $N_2$ and vacuum [130]. The results imply that $N_2$ does not model the space environment quantitatively, and so such an environment may not be predictive in qualification life testing of lubricated components. However, $N_2$ can be used for the purposes listed earlier in this section, and can be especially useful for determining relative cycle life between different coatings using standard testing.

*5.2. Testing Strategies for Flight Parts*

Standard testing is no substitute for conducting tests of the lubricant on parts that are identical to those that will eventually be used in flight hardware. All conditions should be as close as possible to those experienced in flight (i.e., "Test like you fly," or TLYF). For example, the contact motion, speeds, number of cycles, and ambient environment can affect results. It is important to simulate the vibration of launch, which can cause a rocking/fretting motion. As discussed in Section 7.2, this is especially critical for low cycle mechanisms since much of the solid lubricant coating lifetime can be used up during launch vibration. Another effect of launch vibration is that ball bearings can exhibit "gapping," which is a temporary loss of contact between the balls and the raceway, followed by a high contact-stress impact that can cause damage to the bearing [131].

Setting the test requirements is critical in the design process of the lubricated mechanism. Qualification tests are conducted on the spacecraft mechanism to show robustness of the design by choosing a set of operating conditions that are more stressful than expected to be seen on-orbit. Optimally, a life test should be included, where the device must show nominal performance over a specific number of expected lifetimes (often two). Testing vibration at higher levels than launch and thermal cycling over a larger range than seen in space is important, each followed by functional testing.

Acceptance testing of the flight mechanism is usually performed at several levels; first on the component level, then on the subsystem level, and finally at the system-level (i.e., the complete satellite or launch vehicle). The component level test is sometimes deleted, which can be a mistake. If a problem with the lubricant is seen at the component level, subsequent redesign/rebuilding can be significantly less costly than at the subsystem or system stages. The acceptance test may involve operating the

device over some fraction of its intended life on orbit. In addition, thermal cycling and variation of other parameters may be conducted. Such variations are generally more than those seen on orbit to verify robustness, but less than for the qualification tests to avoid impairing the ultimate effectiveness of the device.

*5.3. Qualification by Similarity/Requirement Creep*

Costly failures can occur during advanced stages of manufacture and testing of lubricated parts due to unwitting errors made in the qualification process. Often an application is considered "not too different" from one used previously, so that qualification tests are deemed unnecessary: the part/device is qualified based on similarity to the previous tests/application. Great care must be taken when "qualifying by similarity;" anomalies have occurred during acceptance testing or even on-orbit after underestimating differences in design between two applications. In addition, qualification by similarity cannot be entertained without understanding the failure modes of the mechanism in question. This was highlighted as a potential cause of the slip ring failures occurring on SEASAT [132].

A hypothetical example of qualification by similarity is the use of bearings with self-lubricating retainers used in two different programs under almost identical conditions. The bearings might have the same design, and use the same preload, and have similar speeds during operation. However, seemingly insignificant differences such as differences in ball-to-raceway conformity, level of vibration during launch, reversing motion at regular intervals during operation, or a longer storage period before launch, could create enough performance differences that qualification by similarity is not justified.

Another pitfall can occur when seemingly small changes are made in the design or requirements of a lubricated device after qualification or after previous use on another program. Such "requirement creep" often results in waivers being written to exempt a design or requirement change based on cost. However, such changes may cause anomalous performance. These problems are costlier to correct than the original redesign because extensive failure analysis must often be performed, and because of changes in inflexible delivery and launch schedules. For example, relatively small changes in load can adversely affect performance because solid lubricant coatings tend to have wide life margins, but narrow load margins. The difficulties of requirement creep is discussed with regard to the Thermal Infrared Sensor Instrument (TIRS) in the LandSat-8 program [133].

## 6. Potential Challenges to Successful Application of Tribological Solids

Although solid lubricants solve a number of tribological problems in applications where liquids are not appropriate, their limitations must be considered when designing the lubricated device, and when determining proper test and storage procedures.

*6.1. Humid Air Sensitivity of Solid Lubricants during Storage*

$MoS_2$ is chemically unreactive with respect to most solid and liquid materials. However, its tribological properties may degrade when exposed to humid air due to oxidation of $MoS_2$ to $MoO_3$ during storage [12,134,135]. $MoS_2$ oxidation chemistry is discussed in detail in Ref. [12], and will be summarized here. The predominant degradation mechanism involves chemical reaction of $MoS_2$ with water vapor and oxygen in the air. Two reactions can result:

$$2MoS_2 + 4H_2O + O_2 \rightarrow 2MoO_3 + 4H_2S \tag{1}$$

$$2MoS_2 + 4H_2O + 9O_2 \rightarrow 2MoO_3 + 4H_2SO_4 \tag{2}$$

Both reactions entail $MoS_2$ reacting with $H_2O$ and $O_2$ to produce $MoO_3$, which is not a good lubricating material ($MoO_3$ has been characterized as being abrasive besides being nonlubricating, but this was shown not to be the case [136]). In addition to converting the $MoS_2$ to a nonlubricating form, in reaction (1) the gas $H_2S$ is produced. Although $H_2S$ does not directly affect tribological properties, it can react with silver in $MoS_2$-lubricated slip ring assemblies to produce silver sulfide

(Ag$_2$S), causing electrical noise during operation (see Section 7.3.2). The production of the acid H$_2$SO$_4$ as per reaction (2) has caused the establishment of pH limits of aqueous extracts of MoS$_2$ powders in several specifications, because its presence signifies that lubricant oxidation has occurred, and also because of the concern that it could promote corrosion of steels [46].

Organic-bonded (generally resin-bonded) MoS$_2$ coatings are very resistant to oxidation, since the reactive edges of the MoS$_2$ particles in the coating are covered by the nonporous resin binder, and the resin itself does not absorb appreciable amounts of H$_2$O. The heat-cured coatings are more resistant to moisture incursion, and also to corrosion in general, than are the air-cured coatings. Polyimide-bonded MoS$_2$ coatings are exceptions, since the friction of polyimide increases as it absorbs moisture [18].

Burnished and air-impinged MoS$_2$ and WS$_2$ coatings are formed from powders, and the coatings are somewhat porous, so they exhibit moisture sensitivity greater than that for bonded coatings. However, when burnished MoS$_2$ coatings are used during spacecraft fabrication as an antiseize lubricant for fasteners, their degradation would not impair spacecraft performance.

### 6.1.1. Measurement of Oxidation on MoS$_2$ Powders after Long-Term Humid Air Exposure

Most studies of oxidation of MoS$_2$-based solid lubricants are accelerated by increasing temperature, and also by using high humidities (i.e., >95%). They give relative oxidation sensitivities, but are not useful in predicting how endurance and friction will degrade in a specific spacecraft application. To address this, a long-term, unaccelerated storage study was conducted at The Aerospace Corporation that involved the effect of air with varying humidity levels on stored MoS$_2$ powders [12]. Specifically, the study showed that 3.4 at% of the samples were oxidized to MoO$_3$ after 3 years of storage in air with 53% RH (i.e., relative humidity), while only about 1.3 at% of samples stored at 4–33% RH were oxidized over the same period (see Figure 11). Preliminary oxidation rate data showed that humidities greater than 53% would cause more significant amounts of degradation. As such, considerable improvement in lubricant stability can be gained by reducing the maximum humidity requirement for spacecraft storage from 50% RH, which is commonly-specified, to 35%. (The possibility of damage to spacecraft components due to electrostatic discharge is generally of concern only for relative humidities below 30%; see, for example, the Military Handbook and Military Standard on ESD [137]).

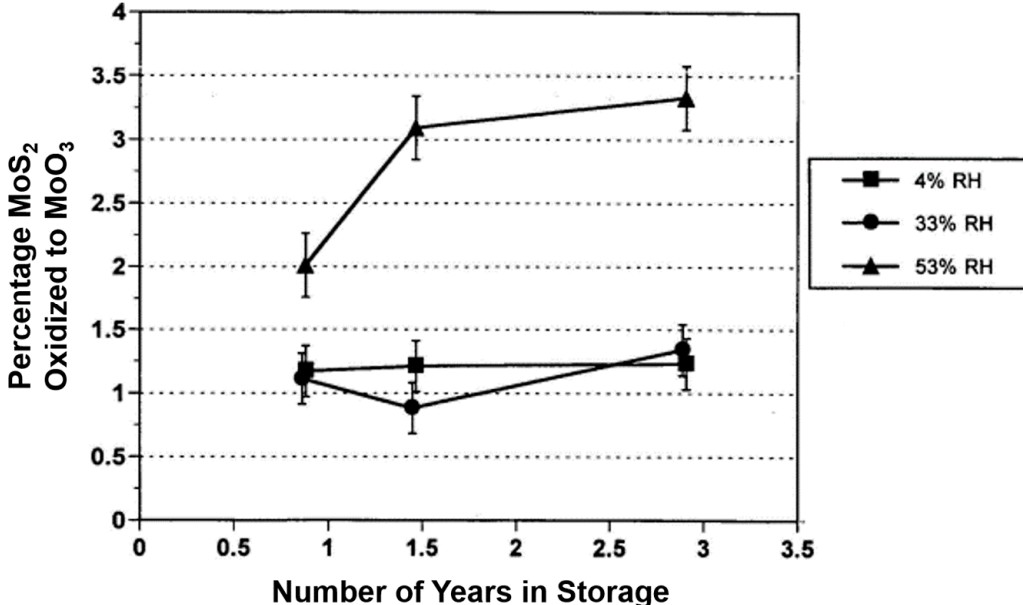

**Figure 11.** Relative amount (in percent) of powder MoS$_2$ samples that were oxidized to MoO$_3$ by exposure to air. Results are shown over 3 years storage for samples stored at three humidities. Reprinted from Lince, J.R.; Loewenthal, S.H.; Clark, C.S. *Wear*, **2019**, *432–433*, 202935, with permission from Elsevier.

### 6.1.2. Tribological Degradation of Sputter-Deposited Nanocomposite MoS$_2$ Coatings after Long Term Humid Air Exposure

Undoped sputter-deposited MoS$_2$ coatings are potentially more susceptible to humidity-induced degradation than MoS$_2$ in burnished or bonded coatings, since they usually exhibit smaller crystallite sizes and greater crystal defect levels. For example, surface chemical studies were conducted of the storage robustness of pure sputter-deposited MoS$_2$ coatings in humid atmospheres [138]. For samples stored in 100% RH air for two weeks, the surfaces showed virtually complete oxidation of the surface. In addition, thrust washer testing of samples stored in 100% RH air for two weeks revealed a decrease in cycle life to 240 cycles, compared to 10,000 cycles for unstored coatings.

Newer coating growth techniques involving codeposition of MoS$_2$ with additives including metals and inorganic compounds are producing coatings with nanocomposite structure. These nanocomposite coatings exhibit greater humid storage robustness and improved tribological properties, due to lower porosity (higher density) and to passivation of the surfaces of the MoS$_2$ crystallites by the additives [80,139]. In addition, this results in a harder, more fracture tough coating that resists wear [69,78].

An unaccelerated storage study was conducted at Aerospace on nanocomposite, sputter-deposited coatings that are commercially available and commonly used on spacecraft [12]. Ni-MoS$_2$, Sb$_2$O$_3$-Au-MoS$_2$, and Sb$_2$O$_3$-MoS$_2$ coatings deposited on 440 C steel were stored at ambient temperature in air with humidity controlled to 59% RH by using a saturated solution of NaBr. POD friction/wear testing of the coated disks was conducted against uncoated 440 C steel balls in a nitrogen gas atmosphere with <0.08% RH. The results were compared to those for samples with no storage time in humid air (see discussion of the performance of as-received, pre-stored samples in Section 3.4).

Endurance data for stored Sb$_2$O$_3$-Au-MoS$_2$ coatings versus storage time are shown in Figure 12. Over the four-year study, the endurance is slightly lower than the unstored coating, i.e., by about 18%. Figure 13 shows corresponding data for the Sb$_2$O$_3$-MoS$_2$ coatings. As for the similar Au-containing coatings, the endurance is lower at the end of the four-year study than for the unstored coating, in this case by about 32%. The endurance degradation of the coatings appears to be greatest during the first year, although the large standard deviation of the data may preclude making such a conclusion. Within the standard deviation of the data, the endurance reduction data for the two Sb$_2$O$_3$-containing coatings over four years are comparable. This would be expected, since the composition of the two coatings is very close: the Au content in the first set of coatings is only 1–2 at%.

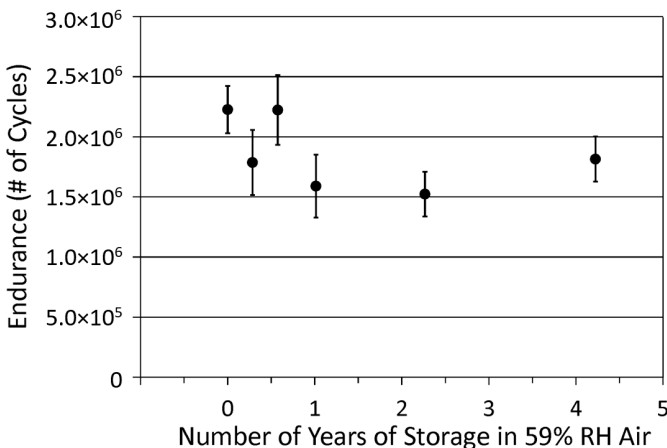

**Figure 12.** Endurance of Hohman Plating Sb$_2$O$_3$-Au-MoS$_2$ coatings as a function of years stored in 59% RH air. The endurance is defined as the number of cycles for which the COF is below 0.5. Each data point represents the mean of three tribometer runs on one sample; the error bars represent the standard deviation of the three runs. The endurance has reduced by approximately 18% for a storage time of 4.2 years. Reprinted from Lince, J.R.; Loewenthal, S.H.; Clark, C.S. *Wear*, **2019**, *432–433*, 202935, with permission from Elsevier.

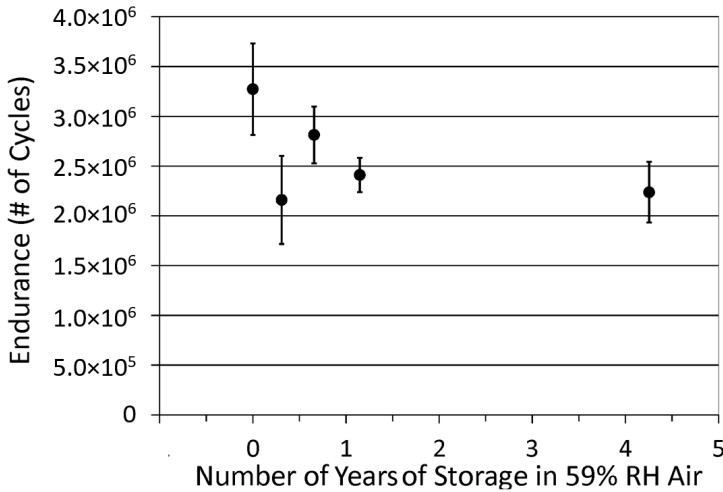

**Figure 13.** Endurance of Hohman Plating $Sb_2O_3$-$MoS_2$ coatings as a function of years stored in 59% RH air. (See Figure 12 caption.) The endurance has reduced by about 32% for a storage time of 4.2 years. Reprinted from Lince, J.R.; Loewenthal, S.H.; Clark, C.S. *Wear*, **2019**, *432–433*, 202935, with permission from Elsevier.

Corresponding endurance data for Ni-$MoS_2$ coatings are shown in Figure 14. In this case, the endurance clearly drops over the first year of storage. It then continues to drop, reaching a total endurance reduction of about 80% over 4.2 years, which represents a significant degradation in performance, considerably larger than for the $Sb_2O_3$-containing coatings.

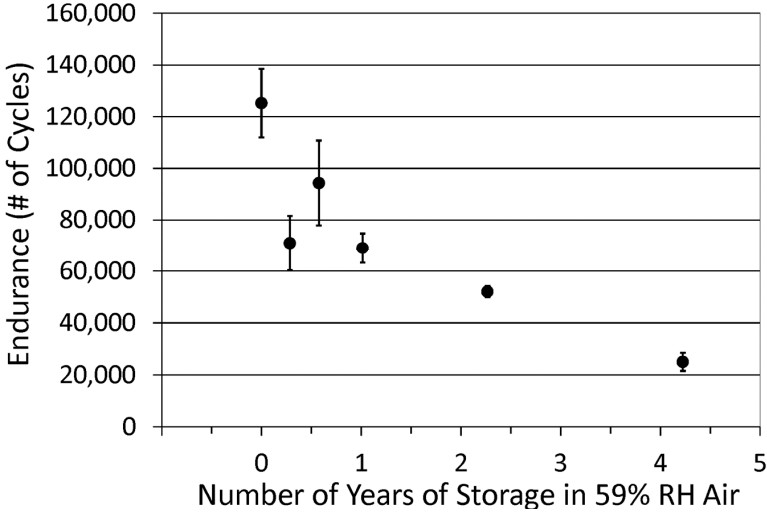

**Figure 14.** Endurance of Hohman Plating Ni-$MoS_2$ coatings as a function of years stored in 59% RH air. (See Figure 12 caption.) There is a reduction in endurance of 80% for a storage time of 4.2 years. Reprinted from Lince, J.R.; Loewenthal, S.H.; Clark, C.S. *Wear*, **2019**, *432–433*, 202935, with permission from Elsevier.

The reduction in endurance during storage was shown to correlate with oxidation of lubricating $MoS_2$ to non-lubricating $MoO_3$ at the surface of the coatings, as measured by X-ray photoelectron spectroscopy (XPS). After storage for 2.3 and 4.2 years, Ni-$MoS_2$ coatings stored in 59% RH air showed significantly greater surface oxidation than a coating stored in dry $N_2$, while $Sb_2O_3$-containing $MoS_2$ coatings showed much smaller increases in oxidation stored under the same conditions [12].

Curiously, curves for all three coatings appear to show an increase in endurance between 0.3 and 0.6 years of storage, followed by a drop in endurance. This interim increase may be related to "maturation" of the coatings due to slow annealing or some other unknown effect. A similar increase

was seen when pure sputtered $MoS_2$ coatings showed an increase in cycle life after 60 days storage in $N_2$ in both sliding and rolling conditions [134].

The results of Ref. [12] are useful as a guide for storing spacecraft hardware lubricated with sputter-deposited $MoS_2$-based coatings. For devices that require a limited number of cycles (i.e., deployment or release mechanisms), oxidative degradation at typical humidities (i.e., <60% RH) will not have a significant effect on performance. However, such degradation must be taken into account for high cycle mechanisms, as endurance can be lowered significantly during storage over several years at typical humidities, depending on the type of coating. Storage in dry $N_2$ or if necessary in air at lower humidities (i.e., <30% RH) would significantly slow the rate of endurance degradation.

In addition to endurance, friction can also be affected by storage in humid air. The XPS results imply that much of the oxidation occurred near the coatings' surfaces. As such, exposure to humid air results in initial increases in friction that may drop after some of the oxidized material is removed, uncovering relatively unoxidized material underneath. During the testing in Ref. [12], the friction was monitored for changes during initial and steady state regimes. For Ni-$MoS_2$ and $Sb_2O_3$-Au-$MoS_2$ stored for 2.3 years in 59% RH air, the maximum COF values measured at the start of the testing were $0.25 \pm 0.005$ and $0.22 \pm 0.01$, respectively. After running long enough to achieve a minimum COF in steady state conditions, the COF values had dropped to $0.007 \pm 0.002$ and $0.003 \pm 0.001$, respectively. Steady state values for samples that were not stored in air were the same for each sample stored in air within experimental error, so although cycle life varied significantly with storage time in air, steady state friction did not.

In another study, air storage of pure sputtered $MoS_2$ coatings was studied [134]. Stored samples showed friction that was higher at the start of the test compared to unstored samples, and steady-state friction values were unaffected by in-air dwell periods (similar to results seen in Ref. [12]). The recovery period was also measured, defined as the number of disc revolutions required to re-establish low friction in vacuum, and this increased logarithmically with storage time.

### 6.1.3. Effect of Moisture Absorption on Inorganic-Bonded Coatings

Inorganic-bonded $MoS_2$ coatings (see Section 3.3.3) exhibit moisture sensitivity, especially when they come in contact with condensed water vapor. The binder, generally a silicate or phosphate, is hygroscopic (can absorb moisture), resulting in weakening the cohesion of the coating. Although inorganic-bonded coatings are preferred for LOX applications (see Section 6.3), their moisture sensitivity can preclude their use in cryogenic LOX applications that involve exposure to air, such as propellant fill valves for launch vehicles, where there are multiple cycles of cooling alternated with returning to ambient temperature.

The moisture sensitivity of silicate binders is highly dependent on their composition, that is, what cations are used (e.g., sodium, potassium, calcium) [140]. Sodium silicate binders are commonly used in solid lubricant coatings used on spacecraft [141], and are particularly susceptible to moisture absorption. This is problematic for several reasons. First, when sodium silicate absorbs moisture, it softens considerably. If the lubricated part undergoes sliding while the coating is softened, then rapid failure could result. However, the moisture absorption is reversible, so if the coating was exposed to moisture in non-operating conditions, and is dried out before use (i.e., in low humidity air or in vacuum), then it may retain its original effectiveness.

Second, beyond softening, sufficient water absorption can result in the formation of an aqueous silicate solution. This can transfer to the opposing surface, forming a "glue" when moisture is removed, so that the two surfaces adhere together. In this case, the effect is not reversible.

Third, after the silicate coatings absorb moisture, the resultant aqueous solution can absorb species from the atmosphere and elsewhere. Of particular concern is the absorption of $CO_2$ from the air in a process known as deliquescence. Hydroxide ion in the silicate solution reacts with $CO_2$ from the air to form soluble carbonate ions. On subsequent drying, sodium carbonate forms as a white

crusty substance in a process known as efflorescence [142]. (It is known in the construction industry as "blooming.")

Bloomed material is nonlubricating, and so can affect tribological performance. Generally, it forms only on the surface of the coating: during initial operation, several cycles of increased friction is seen that then drops down to the nominal friction value for the coating. Tribology notwithstanding, the appearance of a white crusty substance on spacecraft hardware is generally not acceptable aesthetically. In addition, it forms a potential contaminant for nearby optical or mechanical components.

The conditions under which blooming occurs are not quantitatively documented. Generally, softening and resultant $CO_2$ absorption occurs at lower temperatures and higher humidities. A rule of thumb is to keep the humidity below 60% RH and the temperature above 18 °C. Cycling the humidity and temperature above and below the threshold results in increased formation of carbonate material.

Some inorganic-bonded coatings use potassium silicate or aluminum phosphate rather than sodium silicate as the binder. Their friction and endurance also can be affected by storage in humid air, but to a lesser degree due to their lower propensity for absorbing moisture.

### 6.2. Humid Air Sensitivity of $MoS_2$ during Operation

In addition to effects of extended storage in humid air discussed in Section 6.1, exposure of $MoS_2$ to humidity during operation has historically been correlated with degradation of tribological properties [43,143]. For example, in one study, the sliding wear lives of burnished $MoS_2$ coatings sliding on steel were tested in various atmospheres. Wear lives for moist air, dry air and dry argon were in a ratio of about 1:15:150 [144]. Another study evaluated the effect of environment on the performance of a resin-bonded $MoS_2$ coating on an oscillating journal bearing tester, comparing air containing ~50% RH with $10^{-5}$ Pa vacuum [145]. Over a range of contact pressures (i.e., 14 to 84 MPa), the COF for vacuum-tested samples was consistently ~0.05, while that for air-tested samples was ≥0.20. The wear lives for vacuum-tested samples were all one to two orders of magnitude greater than those for the air-tested samples.

The experiences of ESTL on the effect of operating devices lubricated with pure sputtered $MoS_2$ is discussed in detail in Ref. [134] in the current Special Issue, and so will be discussed more briefly here. Running $MoS_2$-lubricated bearings in air under low torque conditions resulted in decreased endurance when subsequently run in vacuum, indicating non-reversible degradation of the coating: in-air running of only 10,000 cycles decreased the life to $0.2 \times 10^6$ cycles, compared to $2.3 \times 10^6$ cycles for bearings not initially run in air. These and previous data are behind the general recommendation to spacecraft contractors not to test $MoS_2$-lubricated devices in humid air.

Since no $N_2$ or even vacuum environments are completely devoid of moisture, this raises the question of to what level humidity should be lowered to avoid deleterious effects on endurance during testing. During POD testing of pure sputtered $MoS_2$, no statistically significant adverse effect of humidity on endurance was seen in the range 0.2 to 2.0% RH; an order of magnitude drop in endurance was seen for samples tested at 45 to 65% RH [134].

It is common knowledge that the COF of sputtered $MoS_2$ coatings (as well as other types of $MoS_2$ coatings) increases significantly in the presence of moisture. In a study of burnished powder $MoS_2$ coatings and sputtered $MoS_2$ coatings, this was shown not to be caused by oxidation, in contrast to that which occurs during long-term storage [146]. Rather, the study showed that at temperatures below 100–250 °C (depending on the type of coating), the frictional sensitivity of $MoS_2$ to humidity is due to water physically bonding to the near-surface and impeding inter-lamellar shear.

The effect of water vapor on inter-lamellar shear is demonstrated by the variation of shear strength with testing atmosphere. A study of sulfur-deficient sputtered $MoS_x$ coatings showed a shear strength of 40 MPa in an inert argon environment, while testing in air with 60% RH gave a shear strength of 80 MPa [29]. Transmission electron microscopy (TEM) along with energy dispersive X-ray (EDX) analysis showed that oxygen was incorporated into the surface film, which was claimed to increase

the chemical interaction between neighboring crystallites. (The techniques used did not ascertain the chemical state of the oxygen within the films.)

In the case of Pb-containing cages and ion-plated Pb coatings, operating such a device in air likely causes oxidation, forming PbO. Initial running in air, followed by operation in vacuum showed an initially higher COF that dropped with continued operation, similar to $MoS_2$ [134].

### 6.3. LOX Compatibility

Organic resin-bonded lubricants are not LOX-compatible because impact of an organic solid in the presence of LOX can result in ignition. In general, inorganic-bonded coatings are used in LOX environments. However, care must be taken to avoid contact between inorganic-bonded coatings and air when they are used below ambient temperature, or water condensation can impair their integrity, as discussed in Section 6.1.3. The compatibility of materials with LOX is detailed in Ref. [147].

### 6.4. Graphite

In order to lubricate effectively, graphite requires a partial pressure of water in the ambient atmosphere. A previous theory held that water is absorbed between the graphite crystal planes near the surface, weakening interplane *p* orbital bonding that allows low friction sliding to take place. This was disproved by *x*-ray diffraction, which showed that the interlayer spacing did not change with water adsorption [148]. The prevailing theory is that water is absorbed at the edges of graphite crystallites, weakening intercrystallite bonding, and allowing sliding to occur on the low surface energy basal plane surfaces [149]. As such, graphite is an excellent lubricant for use at elevations near sea level, but loses its lubricating ability at high altitudes and in the vacuum of space [43,46,150,151]: a partial pressure of water greater than 3 Torr is necessary to reduce the wear rate of graphite to negligible values (~15% RH) [151]. Air has 5–10 Torr water vapor at typical humidities, but the partial pressure of water in space is <$10^{-6}$ Torr. Therefore, graphite in lubricant mixtures is only useful during terrestrial testing in air; even dry nitrogen would impair graphite performance. As a result, most specifications for bonded coatings require that graphite not be used as a lubricating pigment [47,48,152].

### 6.5. Thermal Effects

There have been reports for years that the friction of solid lubricants like $MoS_2$ and PTFE rise significantly at low temperatures. For example, a study in 1976 showed that the friction of unspecified formulations of $MoS_2$ increased by about a factor of two as the temperature was lowered from 250 K to 200 K [153]. This and other results could have been related to testing conditions (i.e., non-vacuum), since the condensation of even a small amount of water on a coating's surface could significantly increase friction. However, more recent low temperature studies conducted in vacuum indicated that this may be an intrinsic materials property. For example, a UHV AFM study of a $Si_3N_4$ probe tip sliding on a single crystal of $MoS_2$ in vacuum showed that the COF increases by at least 30× when the temperature is lowered from 250 K to 200 K [154]. It was proposed that the friction of $MoS_2$ over the temperature range 220–500 K exhibited an activated behavior with an activation barrier of ~0.3 eV. The tip in contact with the surface was described as an interfacial dislocation defect. The motion of the contact exhibited temperature-dependent activated behavior similar to the known temperature dependence of dislocation motion. Below 220 K, the friction behavior became essentially athermal, which measurements showed was associated with the onset of interfacial wear due to the increased energy dissipation.

A recent study was conducted to measure the friction of $MoS_2$-based solid lubricant coatings typical of those used in spacecraft applications at temperatures from 100 K to 300 K in high vacuum conditions (~$1 \times 10^{-8}$ Torr) [155]. The two formulations evaluated were Esnalube 382 ($MoS_2$ with a sodium silicate binder; currently used in spacecraft), and sputter-deposited Au-$MoS_2$ coatings (nanocomposite of amorphous $MoS_2$ and nanosize Au particles; similar to coatings used in spacecraft). As expected, the magnitude of the COF at 300 K is very different for the two coatings. However, both

coatings showed an increase in friction by a factor of ~2 as the temperature was lowered below ~220 K (see Figures 15 and 16).

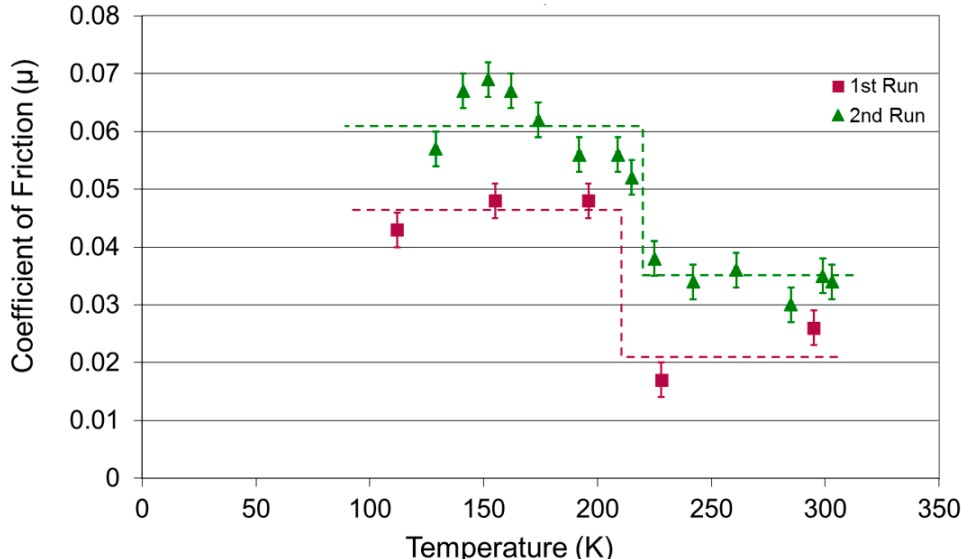

**Figure 15.** Variation of the COF of Esnalube 382 coatings with temperature. Two separate runs are shown. The friction approximately doubles as the temperature is decreased below the transition temperature at ~220 K (−53 °C).

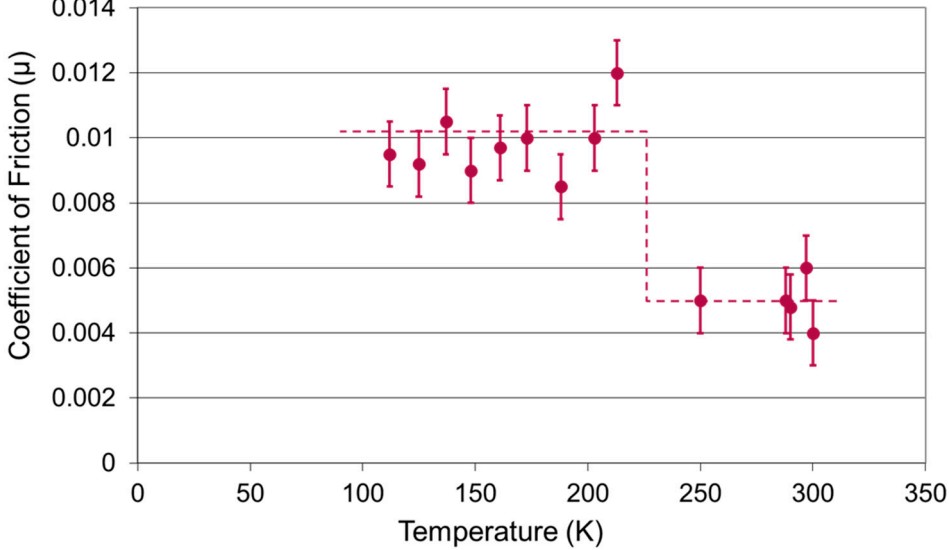

**Figure 16.** Variation of the COF of a sputter-deposited Au-MoS$_2$ coating with temperature. The friction approximately doubles as the temperature is decreased below the transition temperature at ~220 K (−53 °C).

Comparison of the studies on MoS$_2$ solid lubricant coatings [153,155] with that for a nanoscale contact on single crystalline MoS$_2$ [154] indicate that a factor of two increase in friction should be expected when using such coatings in cryogenic mechanisms for space applications.

Studies of PTFE have also shown increased friction values as the temperature was lowered from 400 K (127 °C) to 200 K (−73 °C) [156]. Specifically, the COF increased monotonically with decreasing temperature from 0.075 to 0.210 in a manner modeled well by thermal activation. Small deviations in the smoothness of the curve were found to be correlated with phase- and glass-transitions in the PTFE and temperatures below the frost-points for the respective environments. The overall temperature

variation was modeled by assuming viscoelastic behavior, which varies with temperature via an Arrhenius dependence.

Increasing the temperature above ambient has also been shown to increase the COF of $MoS_2$-based coatings. A study reported in the current Special Issue involved friction force microscopy of a pure sputter-deposited $MoS_2$ coating tested in an inert $N_2$ gas environment. Results showed an increase in both COF and wear volume of 40–50% as the temperature was increased from 25 °C to 120 °C [157]. Ref. [146] saw a similar COF increase over the same temperature range in POD testing of a pure sputter-deposited $MoS_2$ coating in $N_2$; the COF continued to increase as the temperature was raised to 250 °C.

Another issue regarding thermal effects involves the possibility of large temperature excursions during operation. In this situation, the thermal expansion of the coating must be matched with the base material. If they are too different, the coating can fracture and separate from the substrate [18]. This is especially important to consider with bonded coatings, which are much thicker than sputter-deposited coatings.

### 6.6. Atomic Oxygen Exposure

Spacecraft in orbit can be exposed to energetic atomic oxygen; the energy is supplied by the velocity of the spacecraft, so that the atomic oxygen strikes spacecraft surfaces pointing in the direction of the velocity vector (i.e., the "ram" direction). Nearly all materials used in tribological applications will degrade when subjected to atomic oxygen exposure in low earth orbit [92]. However, sealed areas would be immune from degradation, and even surfaces that are shielded from the "ram" direction may be partially or completely protected from atomic oxygen exposure. One study [158] showed that exposing sputter-deposited $MoS_2$, inorganic-bonded $MoS_2$, organic-bonded $MoS_2$, and ion-plated Pb coatings to atomic oxygen beams gave 20–30% increases in initial friction coefficients, although friction dropped, and was the same for irradiated and nonirradiated samples within 2 min after the start of the test. Oxidation depths were 6 nm (0.24 µin) for sputter-deposited $MoS_2$ coatings and 70 nm (2.8 µin) for inorganic-bonded $MoS_2$.

In another study, the effect of atomic oxygen exposure on sputter-deposited $MoS_2$ and diamond-like carbon (DLC) coatings was evaluated. Similar to the study in Ref. [158], exposure of the $MoS_2$ coating prior to sliding showed oxidation only within a few nanometers of the surface, with the result that there was an initial increase in friction that dropped with continued sliding [159]. However, when the coating was continuously exposed to atomic oxygen during sliding, it resulted in a marked decrease in wear life. Exposure of DLC resulted in continuous oxidative gasification of carbon atoms at the DLC surface, which could significantly lower wear life.

### 6.7. Materials Compatibility

Organic-bonded coatings, especially those with phenolic binders, should not generally be used in contact with oils/greases [1,43]. Like organic solvents, hydrocarbon-based lubricants may soften the binder. Similarly, sputter-deposited and ion-plated coatings can be lifted off substrates by the intrusion of oils through defects or grain boundaries in the coatings, resulting in peeling and flaking of the coatings. However, solid lubricants are being used increasingly with oils and greases as discussed in Section 8.5, mostly successfully, although the presence of the coating likely impedes the performance of any antiwear or extreme pressure additives used in the liquids.

### 6.8. Tolerance Budgets

Unlike liquid lubricants, solid lubricants exhibit non-negligible thickness, which should be subtracted from part dimensions when designing bearings and other lubricated devices. Also, manufacturers of bonded lubricant coatings often cannot specify lubricant coating thickness to better than ±0.2 mil (±5 µm), which should be allowed for during calculation of tolerance error budgets.

Design error budgets should also allow for the effect of wear. Even if a lubricant coating thickness is well-characterized, there will be some reduction in thickness over the life of the part, much of which occurs during the early life (e.g., run-in). If wear measurement cannot be made during testing, a post-test surface analysis may be conducted.

### 6.9. Wear Debris

Small amounts of wear are expected to occur for solid lubricant formulations; especially bonded coatings, and early in life. The resultant wear debris must be allowed for when designing spacecraft mechanisms, especially with components that use a hard preload as opposed to a spring preload. Determination should be made of the effect of debris on the performance of the device, as well as potential contamination effects on nearby devices, such as optical sensors.

Solid lubricant wear debris can be a particular problem in low angle bearing operation, where debris pileup at the end of ball travel can result in torque anomalies. Momentary large increases in torque can occur when the oscillatory angle is small, not allowing ball-to-ball overlap before the ends of motion are reached. The result can be the formation of troughs, i.e., solid lubricant wear material piling up at the end of ball travel [160]. There are several ways to deal with this issue. First, the bearing can be designed with smaller balls so as to ensure ball-to-ball overlap. Second, the bearing can be rotated through larger angles with ball-to-ball overlap for a period long enough to achieve a lower wear rate (i.e., via run in). Third, the spacecraft operation can be designed so that larger angle operation (i.e., with ball-to-ball overlap) is cycled with lower angle operation to preclude buildup of wear material at specific bearing angles.

## 7. Typical Applications for Solid Lubricants/Antiwear Coatings in Spacecraft

### 7.1. Actuators

Examples where actuators are used in spacecraft include Solar Array Drive Assemblies (SADAs), deployment devices, and antenna pointing mechanisms. Typically actuators consist of a motor and a gearbox. The support bearings for the actuator and the gears operate at relatively low speeds in the boundary regime, and so solid lubrication is appropriate. The gearbox can be challenging, especially if several reducing stages are being used. The stages may consist of several planetary stages [8], spur gear stages, or a combination of planetary, spur, and worm gears. The tribological requirements can vary between different parts of the gearbox. For example, the input stage (i.e., driven by the motor) operates at a higher speed and lower torque, while the output stage (i.e., that drives the mechanism) operates at lower speed and higher torque. As such, different materials may be required to meet cycle life and load bearing requirements for the gears used in the different stages.

The Bepi Columbo mission makes extensive use of solid lubricated components, since the temperatures may reach as high as 200 °C while in orbit around Mercury [90]. For example, the support bearings and the spur gears for its SADA are lubricated with sputter-deposited $MoS_2$.

The GAIA Space Observatory M2 Mirror (M2M) actuator is using $MoS_2$ to lubricate its ball bearings and gears. To ensure that ground testing did not result in degradation of the $MoS_2$, the testing was conducted in a humidity-controlled chamber [161]. This is in contrast to some manufacturers, who do not control the environment during ground testing, or conduct a series of tests in ambient followed by vacuum. This can result in decreased endurance life and increased torque (at least during initial operation on orbit). However, the missions are successful when care is taken to ensure that the degraded performance still enables meeting requirements with adequate margin.

The three Focus and Alignment Mechanisms (FAM) in the Near Infrared Camera (NIRCam) on JWST are linear actuators required to provide micron level positioning in tip, tilt and piston to the Pickoff Mirror that reflects the starlight into the rest of the NIRCam instrument. The FAM, along with the other NIRCam mechanisms such as the filter wheel assembly and pupil imaging assembly, are required to operate at temperatures of ~37 K for near infrared imaging during the 5 year JWST mission.

Such low temperatures require solid lubrication. The sliding surfaces in the FAM are being lubricated with nanocomposite sputter-deposited $Sb_2O_3$-$MoS_2$ coatings (see Figure 17) [13].

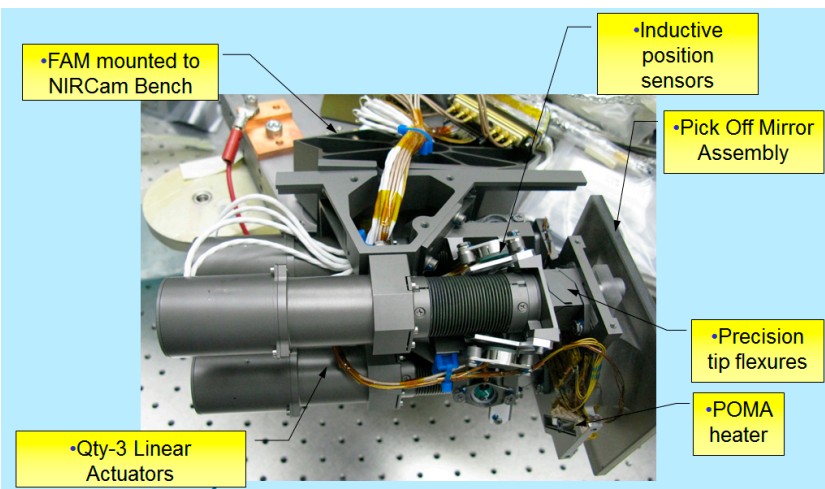

**Figure 17.** The Focus and Alignment Mechanism (FAM) provides tip, tilt and piston for NIRCam's Pick-off Mirror (POM). Reprinted from Lince, J.R.; Loewenthal, S.H.; Clark, C.S. Proceedings of the 43rd Aerospace Mechanisms Symposium; NASA Conf. Publ. NASA/CP-2016-219090, 4–6 May 2016.

### 7.2. Deployment and Release Mechanisms

Mechanisms used during deployment generally require limited cycles (or only one) on orbit. As such, it is often thought that the tribological requirements are minimal. As such, in new programs, inadequate attention may be given to the materials used, including both the base materials and low friction coatings. All good size satellites include shear ties that ensure structural integrity of solar arrays during launch vibration, latches/locking mechanisms, and ball lock mechanisms. Pyrotechnic release mechanisms like bolt cutters, wire cutters, and pin pullers often have multiple parts lubricated with solids. Bonded solid lubricant coatings based on $MoS_2$ are often used for these devices.

The interface for many of these devices is often a cup/cone or sphere-on-a-sphere arrangement (see Figure 10). However, rarely are these area contacts: they are often line contacts (around the cup, cone, or sphere circumference), and so generally exhibit significant Hertzian contact stress ($S_{max}$) values, i.e., in the tens of thousands of psi. Launch vibration can cause a rocking/fretting or impacting contact [162]. So even though these devices are used minimally on orbit, fretting due to vibration can increase the total effective lifetime to five or ten thousand cycles. As such, robust vibration testing should be done evaluating different solid lubricant/antiwear coatings, followed by TLYF functional testing and visual examination [163].

Another way of increasing robustness for release devices based on cup/cone or sphere/sphere contacts is to optimize the contact angle, defined as the angle between the axial centerline and the line of contact tangency between the cup and cone (or sphere and sphere). Increasing contact angles result in increasing the critical friction ($\mu_c$), which is the COF above which the cup/cone will not release. Increasing $\mu_c$ will make the device more tolerant of increased friction due to anomalous galling, but must be balanced against the propensity for shear motion to increase at high contact angles.

### 7.3. Solid-Lubricated Slip Ring Assemblies

It is often necessary to pass electrical signals and power between different subsystems on the spacecraft. This is accomplished using a slip ring assembly (SRA), which comprises a series of sliding electrical contacts between stationary brushes (in the form of single wires, bundles of wires, or composite blocks) and rings (precious metal alloy rings, either solid or plated). There are three main applications for SRAs [6]. The main application is passing electrical power from solar arrays to the

satellite bus. In addition, satellites with control moment gyroscopes (CMGs) need to pass electrical power from the spacecraft bus through the motorized gimbals to the gyroscope motors. The last major application is passing signal and power to rotating microwave sensors on weather satellites and other earth-observing satellites.

The main types of SRAs are shown in Figure 18. They are all based on a balance between low friction and wear on the one hand, and low electrical contact resistance on the other; this is challenging because both solid and liquid lubricants tend to be poor electrical conductors. Solid lubricated SRAs are based on composite brushes comprised of a conductive metal like Ag, Au, and/or Cu, mixed with $MoS_2$. The brushes are contoured to fit the OD of a cylindrically shaped slip ring, generally made of a silver alloy material or coating. Such self-lubricating slip ring brushes have been used since the late 1960s on spacecraft [164]. Monofilament SRAs use a single wire for the brush that rides in a V- or U-shaped channel on the slip ring. Both brush wire and ring are comprised of noble metal alloys, and are oil-lubricated [6,165]. The oil operates in the boundary regime to allow electrical conduction from brush to ring, while also reducing wear to acceptable levels. Historically, these two technologies have comprised the majority of SRA applications on spacecraft.

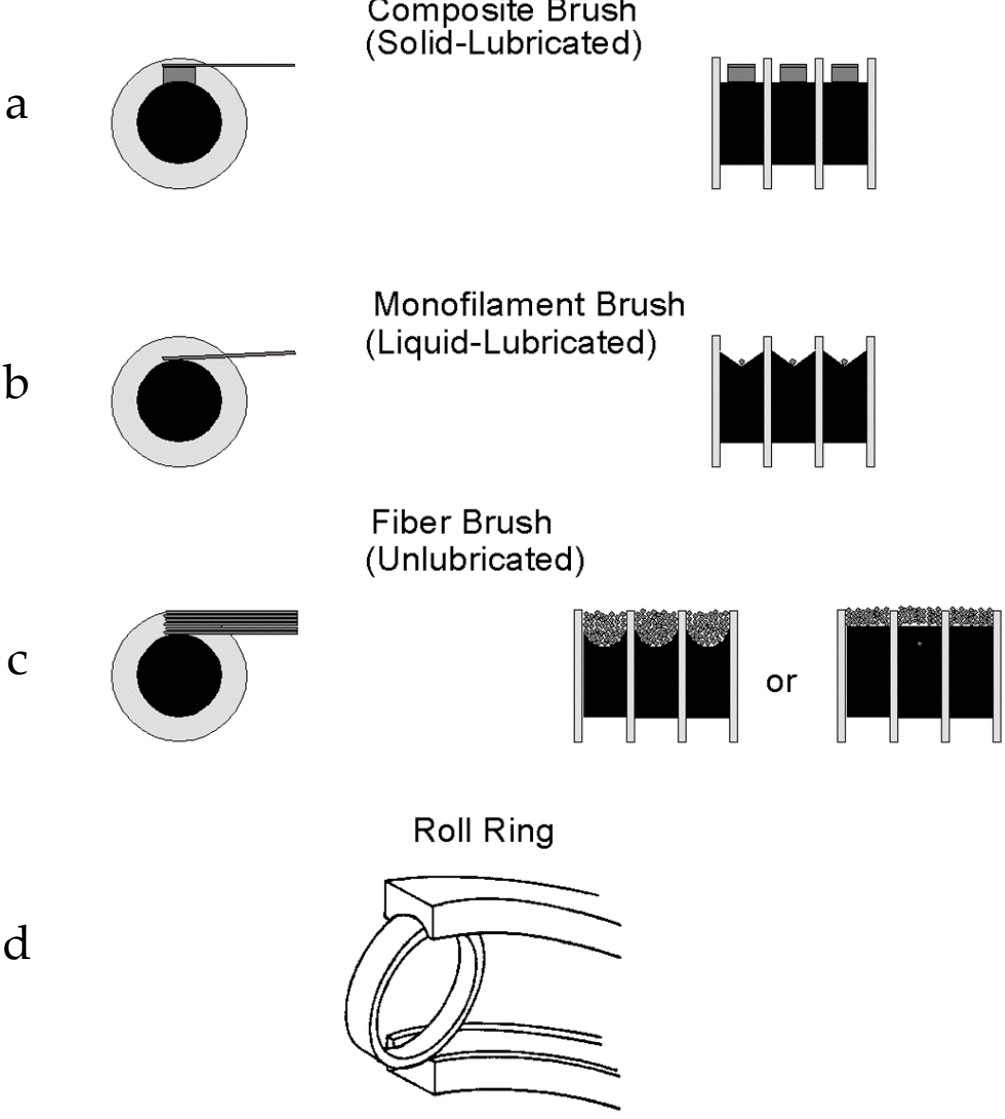

**Figure 18.** Cross sections of several slip ring technologies, including (**a**) composite brush (solid lubricated), (**b**) monofilament brush (liquid lubricated), (**c**) fiber brush, and (**d**) roll ring.

More recently, the fiber brush technology has been developed that replaces monofilament brushes with fiber brushes that each contain many filaments [6,166]. Because of the reduced $S_{max}$ per fiber compared to the monofilament design, these are often used without lubricant. SRAs based on fiber brushes are experiencing increased application on spacecraft. The roll ring assembly (RRA) is based on the design of rolling element bearings, using conductive flexures rather than solid balls; this technology has been used on the ISS [6,95], but has shown limited application otherwise.

Solid-lubricated, monofilament, and fiber brush SRA technologies each have strengths and weaknesses. Solid-lubricated designs are versatile in that they can operate at a range of speeds, carry significant amounts of current, and last for many years. A significant issue is that they are sensitive to exposure to humid air during storage and ground testing, which will be discussed below.

Monofilament designs are also versatile in that they have long lifetimes, are less sensitive to air exposure (due to the precious metal materials and coverage with oil), easier to manufacture, and more slip ring circuits can be included in a SRA (due to smaller ring-to-ring axial pitch). However, because they are oil-lubricated, using at higher speeds and lower temperatures can result in significant electrical noise. An example is the monofilament slip ring design used in Windsat, a higher-speed microwave imager program that showed electrical noise at temperatures below ambient [167].

The fiber brush technology avoids some atmosphere sensitivity and speed effects, and can be used over a range of temperatures. Because of the lack of lubrication, they may have lower cycle lifetimes than the first two technologies; these are being used extensively for slower mechanisms like SADAs.

The solid-lubricated SRA is a special case of solid tribological materials, and is a critical subsystem used on many spacecraft, and so will be discussed in some detail here.

As discussed in Section 7.1, the Bepi Columbo mission to Mercury requires solid lubricated components, since the temperatures may reach as high as 200 °C. As such, they chose a solid-based ring/brush arrangement, with brushes made of an $Ag/MoS_2$-based material, even though the rotation speeds are likely low [168].

7.3.1. Fabrication and Testing of Solid-Lubricated Slip Ring Assemblies

Currently, there are two formulations of silver/$MoS_2$ SRA brushes in wide usage on spacecraft. The first is used primarily in U.S. spacecraft, and contains graphite, to give a final composition of 85 wt% silver, 12 wt% $MoS_2$, and 3 wt% graphite [169,170]. (The graphite was originally added to provide lubrication for high altitude aircraft, which required graphite lubrication in humid air at low altitude, and $MoS_2$ lubrication in low humidity at high altitude). The second formulation is that originally proposed by Clauss and Kingery [171], and used by the European Space Agency (ESA) and other countries [7,172]. This formulation contains copper to increase hardness, with typical composition in the range of 82.5 to 85 wt% silver, 12.5 to 15 wt% $MoS_2$, and 2.5 wt% copper. The brushes are fabricated by hot-pressing powders together. The brushes are sintered; then one surface is contoured to the shape of the SRA ring by grinding and polishing on mandrills with the same diameter as the ring.

The brush is soldered to a cantilever formed of a metal strip with an appropriate spring constant (usually a copper-beryllium alloy). The soldering process must be controlled to avoid excessive oxidation of the brush. Brush pressures are generally in the range of 4 to 30 psi, with 10 to 20 psi being typical, representing a trade-off between wear and electrical noise, since lower pressures result in lower wear, but noise usually increases at pressures under some critical value [173]. Typical current densities for solid-lubricated brushes are below 50 A/in$^2$ for signal circuits and in the range of 100 to 600 A/in$^2$ for power circuits. There is no theoretical limit for rotation speeds, although they are routinely operated at 30 to 60 rpm.

The rings used with silver/$MoS_2$ brushes are generally high silver alloys. Two technologies are generally used. The first is a solid piece of coin silver (90 wt% silver and 10 wt% copper), where copper is present as a hardening agent. The second ring technology relies on a plating process, whereby a copper or phenolic cylindrical substrate is first plated with one or more layers of metal, including copper and/or silver. The final layer is a silver-based alloy, often consisting of a "hard silver," which is a

high silver alloy with other minor constituents. The ring rotor assembly (i.e., the SRA minus the brush block assembly) is formed by mounting the rings consecutively along the rotation axis. Each ring is separated from the next by a barrier whose diameter is larger than the ring diameter. The difference between the barrier and ring radii should be greater than the brush height to minimize the chance that brush/ring debris could short out adjacent ring circuits (see Figure 18).

After the solid-lubricated SRA is assembled, and the brushes are in contact with the rings, it is important to minimize operation in air. This is because of the oxidation sensitivity of SRAs, which is discussed in Section 7.3.2. Some SRA manufacturers may recommend initial operation for a short time in air to encourage initial formation of a $MoS_2$ transfer film on the ring. However, it has been known for many years that significant operation should be conducted in vacuum [174]. If necessary, short-term testing (i.e., for minutes, not hours, and especially not days) can be done in an atmosphere consisting of dry filtered nitrogen gas.

To help ensure that a flight SRA will behave optimally on orbit, extended non-accelerated testing (i.e., for at least a week) may be conducted in an appropriate vacuum test chamber. This testing greatly reduces the risk of "infant mortality." In addition, brush/ring wear rates are high during initial operation. As such, the debris can be removed after testing to lessen the possibility of the SRA shorting on orbit; the unit can be designed to enable access to the interior of the SRA for cleaning by vacuum, for example. Also, removed debris can be examined to ensure anomalous wear conditions do not exist.

7.3.2. Atmospheric Degradation during Storage/Testing

Most issues with using solid-lubricated SRAs are related to the oxidation of $MoS_2$ in the brushes. As discussed in Section 6.1, exposure of $MoS_2$ to humid air at room temperature can result in oxidation to $MoO_3$. Oxidation during terrestrial storage can contribute to anomalous ring and brush wear. Because $MoO_3$ is electrically insulating, it can cause increases in electrical resistance; it has been associated with high-frequency noise of very short duration (1–20 $\mu$sec in width).

Lower frequency noise sometimes occurs with solid-lubricated SRAs, where high resistance is seen at a particular location on the slip ring in slow-moving or even static conditions. This is likely due to the production of $H_2S$ gas during the oxidation process [see Reaction (1)]. Silver (Ag) reacts strongly with $H_2S$ to form $Ag_2S$ via the following reaction:

$$2Ag + H_2S + \frac{1}{2} O_2 \rightarrow Ag_2S + H_2O \tag{3}$$

For example, at <5% RH, an approximately 4 nm $Ag_2S$ film is formed on silver at room temperature after an $H_2S$ exposure of only 100 ppm-hr [175]. At >75% RH, a film greater than 30 nm thick is formed.

$Ag_2S$ is a semiconductor, and therefore can cause electrical noise when it forms a thin film on a slip ring surface, interrupting conduction between the slip ring and the brush. This agrees with the common observation that high resistance anomalies often occur in the null position on spacecraft SRAs, i.e., the relative brush/ring position fixed during storage, launch, and orbital insertion. The $Ag_2S$ film would be expected to grow predominantly at the position directly under the brush, the source of the $H_2S$ gas.

The formation of a high resistivity $Ag_2S$ film is not chemically reversible. Once it has formed on the surface of a slip ring, it may be necessary to operate an SRA for tens of thousands of revolutions in vacuum to wear away the $Ag_2S$ film. High slip ring resistance is often seen when SRA-containing subsystems are first operated on orbit; the thin $Ag_2S$ film usually disappears quickly during initial operation. However, more severe problems involving persistence of high resistance or electrical noise have occurred that may be due to problems involving extensive atmospheric exposure and/or brush processing variations. Examples include the ESA Marecs A satellite [176] and DSCS III [173].

The oxidation mechanism shown in reaction (1) involves reaction of $MoS_2$ with water and oxygen. This observation, and the results of Ref. [175], indicate that an SRA should be placed in a dry nitrogen gas (or other inert) atmosphere during storage to limit exposure to both water vapor and oxygen.

SRAs can be fitted with connections to allow continuous purging with nitrogen gas. This ensures the optimum atmosphere for the brushes/rings whether or not the overall spacecraft is stored in air. (This arrangement also allows purging the SRA to within a few days of launch.) Alternatively, the SRA, the subsystem containing the SRA, or even the entire spacecraft could be placed in a sealed container containing nitrogen and desiccant, or could be continuously purged with nitrogen if the container is difficult to adequately seal. There is a legitimate concern that low humidity enhances the possibility of electrostatic discharge (ESD), which could damage sensitive components. However, such concern should be mitigated by grounding all components during storage and using other anti-ESD procedures (see for example, Ref. [137]).

The limitation of water and oxygen is especially important during testing, when the adverse chemical reactions may be enhanced by increased brush/ring contact temperature or tribochemistry. Ideally, an SRA should be operated only in vacuum, although logistics may require performing limited operation in a dry nitrogen atmosphere.

### 7.4. Other Applications

Launch vehicles are a special application for solid lubricated ball bearings. In particular, LH2/LOX-fueled launch stages have bearings in high speed turbopumps that cannot be used with liquid lubricants, both because they cannot be used with the wide temperature ranges involved in pumping cryogenic liquids, and because hydrocarbons are not LOX compatible (i.e., are a combustion risk). A variety of self-lubricating composites have been put to use in bearings in the Space Shuttle Main Engine (SSME) and many upper stage engines (and some lower stages). The SSME originally utilized 440 C bearings whose cages were made of a fiberglass cloth that was impregnated with PTFE. The lifetime of these bearings were short (the fiberglass fibers wore against the balls, increasing frictional heating), and after a test program were replaced with hybrid bearings with silicon nitride balls that contained cages made of a PTFE/Bronze composite [177,178].

Attitude control devices such as RWs and CMGs on large satellites are typically lubricated with oils and greases. Solid lubricants are not used because of the perceived limited lifetime and the potential for torque noise due to solid lubricant debris. However, solid-lubricated reaction wheels have been used successfully for smaller satellites. Bearings in SSTL satellites generally use PGM-HT bearing cages, and rely on transfer of the $MoS_2$ lubricant to the balls and races [10]. The bearings for many SSTL products can be operated at <300 rpm, which maximizes life, but have been used at speeds >5000 rpm successfully. One advantage of solid-lubricated bearings is that they can undergo accelerated life testing, which is non-conservative for liquid-lubricated bearings. For the Giove-A program, the short development time necessitated an accelerated testing program, enabled by using solid-lubricated bearings.

Another atypical application of solid lubrication is the drive unit for the hammering mechanism for the HP3 mole-type penetrator, used for digging holes on Mars for NASA's InSight mission. The internal parts were extensively lubricated using $MoS_2$ and Pb, deposited on top of hardened metal surfaces [179].

## 8. Future of Tribological Solids on Spacecraft

### 8.1. Highly Hydrogenated Diamond-Like Carbon (HH-DLC)

Solid lubricants generally have low shear strength and low surface reactivity, which is the basis for their low friction, but also gives rise to a non-negligible wear rate, which limits their cycle life. The Holy Grail in tribological spacecraft materials is a low friction solid with high hardness: it would achieve the low friction of materials like $MoS_2$ and PTFE, but exhibit a much lower wear rate. DLC and other hard carbon-based coatings show promise for filling this niche, but the coatings typically used in terrestrial applications do not show promise in the vacuum of space. Like graphite, they require

some ambient humidity to provide low friction [180] and wear [181], because they contain little or no hydrogen.

In contrast, highly hydrogenated DLC (HH-DLC; >40% H) exhibits significantly different behavior, specifically very low friction and wear in the absence of water vapor [180]. Hydrogen atoms bond strongly to carbon atoms, effectively passivating their unoccupied or free σ-bonds. Such passivated carbon atoms show little adhesive interactions during sliding, with resultant low friction. In illustration, hydrogen provides low friction when adsorbed on the surface of bulk diamond or diamond films, and under conditions where the hydrogen is removed, friction increases dramatically. With HH-DLC, in addition to surface-adsorbed hydrogen, there is a reservoir of hydrogen in the bulk, which can replenish the surface if it is removed, maintaining good tribological properties [180].

In a POD tribometer study with both disks and balls coated, the coatings exhibited superlow friction (COF < 0.004) at UHV, and for $H_2O$ pressures up to 13 Pa (0.1 Torr) [182]. As the $H_2O$ pressure rises, so does the COF, settling at a value of 0.07 at 1333 Pa (10 Torr), as well as a similar value in air with the same $H_2O$ partial pressure (i.e., corresponding to ~50% RH). The study also showed that the friction was reversible: after testing with $H_2O$ and $O_2$, ultralow friction was again achieved after vacuum was reapplied. In addition, SEM observation showed negligible wear had occurred: only light burnishing marks were seen due to the removal of asperities in the coating. Extremely low wear has been confirmed quantitatively in Ref. [180], where wear rates in the range of $10^{-9}$–$10^{-10}$ $m^3/N·m$ were measured in dry $N_2$; wear rates were inversely related to hydrogen content.

Although there have been relatively few studies of the HH-DLC coatings involving testing in space applications, they have shown significant promise. A study was conducted of angular contact bearings with HH-DLC-coated races and steel cages, and uncoated balls [183]. The torque value in vacuum was similar to that for $MoS_2$-coated bearings, and the cycle life was twice that for $MoS_2$. Relatively little wear of the coatings were seen. Another study tested a thrust bearing in vacuum where the races, cages, and balls were coated with HH-DLC [184]. The bearing survived for 1.7 million cycles, and at lower torque than for uncoated bearings, which failed almost immediately (i.e., <10,000 cycles).

Bearings coated with HH-DLC could be an advantage when used with liquid lubricants. Bearing failure is usually caused by lubricant loss or degradation. HH-DLC coated steel parts preclude metal-to-metal contact, potentially reducing lubricant degradation. In addition, should the lubricant be depleted, the HH-DLC-coatings could forestall failure by providing continued low friction for a time. However, liquid lubricants might need to be reformulated to be used with HH-DLC, since common spacecraft lubricant antiwear additives are chosen because their tribochemistry is optimized for use with steel. In fact, additive tribochemistry even varies with type of DLC coating. For example, one study found that some typical terrestrial lubricant additives worked better with hydrogenated DLC coatings, while others worked better with non-hydrogenated DLC coatings [185].

To this effect, a study of thrust bearings with HH-DLC-coated races were tested in vacuum that were lubricated with multiply-alkylated cyclopentane oil (MAC—also known as Pennzane—a common spacecraft oil) [186]. The oils were unformulated, or formulated with either aryl phosphate esters or lead naphthenate. The bearings lubricated with either formulated oil and coated with HH-DLC showed minimal improvement in cycle life over similarly-lubricated uncoated bearings. This is consistent with measurements by surface chemical analysis that showed virtually no additive-containing tribofilms on the coating surfaces. For bearings lubricated with unformulated MAC oil, the HH-DLC-coated bearings showed statistically significant endurance improvement over the uncoated bearings, likely due to prevention of metal-to-metal contact.

A similar study of HH-DLC-coated thrust bearings was conducted, but using Braycote 815Z oil, a perfluoro-polyalkylether (PFPE) [184]. Braycote 815Z is known as having a very low vapor pressure, but can show lower life than MAC oils when used in boundary conditions and for somewhat over a million stress cycles, due to tribochemical interaction of the oil with steel surfaces [187]. The HH-DLC-coated bearings showed cycle lives of 30–50 million cycles, while that for uncoated bearings

was ≤10 million cycles. The improvement was ascribed at least partly to reduced chemical degradation of the oil due to the relative inertness of DLC compared to the steel surface.

The results on HH-DLC-coated ball bearings are clearly promising for spacecraft applications, but no such application has of yet been effected. One reason for this lapse is that spacecraft manufacturers and their customers are often resistant to change, hoping to avoid "catastrophic improvement." Another reason is that there is no terrestrial market for these coatings, and so no manufacturing outside of laboratories exists.

Technical issues that must be addressed to achieve insertion of HH-DLC into spacecraft mechanisms include standardization of the coating growth process, correlation of the actual spacecraft environment with laboratory environments, investigation of fretting behavior (i.e., to model the effect of launch vibration), and further study of the interaction of spacecraft lubricants (and their additives) with HH-DLC-coated parts [188].

### 8.2. Cubic Boron Nitride (c-BN)

Cubic boron nitride (c-BN) is another promising hard coating for spacecraft use, because of its low friction in vacuum in some applications [189]. Specifically, sliding COF values of ~0.01 were seen in vacuum, and 0.04 in dry $N_2$ for c-BN sliding against pins coated with CVD diamond, with relatively low wear rates of ~$10^{-6}$ and ~$10^{-8}$ $m^3$/N·m, respectively. However, when sliding against uncoated metal pins in vacuum, the COF becomes higher: 0.8 against steel and 0.4 against aluminum [190]. Although these results suggest c-BN for vacuum and spacecraft use in limited applications, it should be noted that HH-DLC sliding against itself exhibits an order of magnitude lower friction than c-BN, and two orders of magnitude lower wear rate in dry $N_2$ (0.004 and $10^{-10}$ $m^3$/N·m, see Section 8.1). However, the hardness of c-BN is 2–10 times greater than for the HH-DLC coatings (depending on hydrogen content), so c-BN coatings may be preferred for that reason in some niche applications [191,192]. In addition, c-BN is highly refractory, and so may be able to be used in higher temperature applications.

### 8.3. Surface Microtexturing

Laser microtexturing has been performed to improve tribological performance of surfaces used with both liquid and solid lubricants, and although promising, has not yet been applied to spacecraft hardware. Microtexturing results in dimples or troughs that can act as reservoirs, resulting in transfer of solid lubricant material into the contact region at an optimum rate. In addition, the dimples or troughs can trap wear particles, moving them below the contacting plane.

Laser microtexturing has been shown to significantly lower friction and increase cycle life. An example is microtextured M50 steel lubricated with a composite made of $MoS_2$ and polyimide powders [193], which showed friction reduction and life increase compared to non-textured surfaces lubricated with the same material. In addition, microtextured TiCN surfaces lubricated with burnished $MoS_2$, sputtered $MoS_2$ or $MoS_2$/graphite/$Sb_2O_3$ powders were studied [194]. Microtexturing produced significant wear life increases for the burnished and sputtered $MoS_2$ coatings.

### 8.4. Adaptive "Chameleon" Lubrication

The combination of $MoS_2$, graphite, and $Sb_2O_3$ powders in Section 8.3 is an example of adaptive "chameleon" lubrication [195], where one species lubricates effectively in vacuum ($MoS_2$), while another lubricates better in humid air environments (graphite). In that study, testing environment, COF, and surface chemistry of the microtextured coating were studied during POD testing against a steel ball. When testing in humid air (40% RH), Raman spectroscopy in the wear track revealed that graphite was enhanced in the transfer film, while in dry $N_2$, $MoS_2$ was enhanced in the transfer film. As such, the optimum solid lubricant was present in the contact region for each environment [194].

Initial adaptive coatings were W-C-S mixtures produced using magnetron assisted pulsed laser deposition (MSPLD) to obtain WC/DLC/$MoS_2$ nanocomposites [196]. The WC provided a hard matrix, while the DLC provided low friction/wear in humid environments, and $MoS_2$ provided low

friction/wear in dry and vacuum environments. The surface enhancement of each lubricating material was caused by rapid wear of the other material in the adverse environment, i.e., wear of $MoS_2$ in humid environments, and wear of the DLC in dry/vacuum environments. The utility of this type of system is enhanced by its ability to cycle between environments multiple times while retaining good tribological performance.

Higher temperature performance has been achieved with a mixture of yttria stabilized zirconia (YSZ) in a gold matrix with encapsulated nano-sized reservoirs of $MoS_2$ and DLC [197]. The YSZ provided a hard nanocrystalline matrix, the $MoS_2$ and DLC provided lubrication at room temperature in dry and moist environments, and the gold provided lubrication at elevated temperatures <500 °C. Although it was shown that the system worked well during environment cycling at ambient temperatures, operation at elevated temperatures could degrade the DLC and $MoS_2$, precluding multiple temperature cycles.

Further studies were conducted with YSZ-Ag-Mo composites tested over a large temperature range: 25–700 °C [198]. They were found to exhibit a moderate friction coefficient of 0.4 over the full range of temperatures. The adaptive quality of the coatings was demonstrated by surface analysis that revealed that the contact surfaces were composed of Ag from 25 °C to 500 °C, and of $MoO_3$ at temperatures above 500 °C. $MoO_3$ is one of several oxides that are known to form Magnéli phases, which possess high-defect low-shear crystallographic planes in high temperature oxygen environments, with resultant good tribological properties [199]. Other metals that form Magnéli oxides that could be useful high temperature lubricants include W, V, Ti, and Re [195].

Some hard metal nitride coatings may also exhibit adaptivity in high temperature oxidizing environments by forming lubricious metal oxides on their surfaces. For example, VN coatings begin forming lubricious $V_2O_5$ on their surfaces as the temperature is increased above 400 °C. As a result, the COF goes from 0.45 at room temperature to 0.25 at 700 °C [200].

Refractory ceramic coatings may exhibit reduced friction at moderate temperatures (i.e., <500 °C) by mixing with low shear strength metals. For example, carbides and nitrides like TiC, TiN, and CrN have been used to form composites with Ag. COF values in the range 0.2-0.4 for 10–20 at% Ag were seen in the range 25–500 °C. Issues with these materials include higher wear rates than the pure ceramic, and relatively rapid depletion of the Ag from the ceramic matrix at the higher end of the temperature range [195].

Premature oxidation of metals in the coatings, and depletion of the lubricating metal (i.e., Au or Ag) are two issues that limit reversible adaptive behavior during temperature cycling. One proposed solution is the use of diffusion barriers. This was tested for YSZ-Ag-Mo coatings with the use of TiN diffusion barrier layers: the TiN layers had patterned holes to meter the Ag diffusion rate to the surface of the coating. As a result, the coating endurance was found to be more than 10× that for a coating without TiN [201].

Continued research in materials adaptive to varying environments and elevated temperatures would be especially useful for a new generation of vehicles that operate in air while transiting to space, and capable of multiple atmospheric reentries. On spacecraft, these materials can be useful when solid-lubricated devices must be tested extensively in humid air prior to launch. However, the improved tribological properties of these materials in air does not absolve the spacecraft engineer from practicing TLYF, ensuring that the coating will act as intended in the spacecraft environment after ground testing in air.

### 8.5. Hybrid Liquid/Solid Lubrication

Parts coated with solid lubricant coatings like sputter-deposited- and bonded-$MoS_2$ have begun to be used along with oils and greases. Often, pefluoro-polyalkyl ethers (PFPEs) like Braycote 601 grease are used with devices coated with sputter-deposited $MoS_2$ [202]. There are two benefits to this approach. First, the grease provides some protection against oxidation to the coating during storage

and testing in atmosphere. Second, in boundary applications, the solid coatings provide an effective means to avoid metal-to-metal contact and resulting tribochemical degradation of the PFPE lubricant.

Ref. [202] showed mixed results when using such a hybrid system. The study showed improved cycle life in boundary rolling conditions (i.e., SOT testing [203]) when using sputtered $MoS_2$ coatings along with Fomblin Z25 oil (a PFPE), compared to using either separately. For sliding conditions (i.e., POD), a hybrid sputtered $MoS_2$/Braycote 601 combination exhibited higher life than the grease alone. However, the hybrid showed lower life than the $MoS_2$ alone (except for higher $S_{max}$ and elevated temperature). When the test was repeated using Fomblin Z25, a modest improvement was seen compared to $MoS_2$ alone. The results indicate that using a hybrid system will be advantageous under conditions when the critical $MoS_2$ transfer film is allowed to form. If the liquid lubricant removes the $MoS_2$ third body from the contact region, the wear rate of the coating will be higher than if the liquid was not present.

In some cases, a grease is added after preflight testing when it appears that the surfaces of parts lubricated with bonded solid lubricant coatings have been damaged; the grease acts as a confidence builder, even when the visually anomalous coating appearance does not result in test failure.

## 9. Conclusions

Tribological solids like dry film lubricants, self-lubricating composites, and hard antiwear coatings will continue to make critical contributions to spacecraft design. Liquid lubricants still cannot be replaced for such applications as devices operating at high speeds and with numerous cycles such as spin bearings. However, solids have filled many niches where containment of liquids is a problem, both from a contamination and a lubrication standpoint, and applications running in the boundary regime. Examples of such niches include deployment mechanisms, actuators, solar array drive bearings, structural connections (rivets and screws), and slip ring assemblies. In addition, the use of solids to provide lubrication is critical for applications involving temperature extremes, such as scanner and gimbal bearings used with low-temperature infrared sensors, cryogenic valves used on propellant tanks, and turbo bearings in LH2/LOX fueled rocket engines.

Even though solid lubricants are ubiquitous on spacecraft, their use could be more widespread if there was a greater understanding of their strengths and Achilles' heels. For example, $MoS_2$ has a well-documented sensitivity to humid air that has soured some people on its use, but the degree of performance degradation depends on the formulation, specific exposure history, and the intended use. In illustration, this review showed that the effect of humid air exposure during storage had very different effects on different types of sputter-deposited $MoS_2$-based coatings.

Some failures have resulted because the type of solid lubricant coating was chosen after all materials and designs had already been finalized. However, the choice of the correct lubricant formulation and the development of successful lubricant application methods should be integral parts of the design of a lubricated device. A main goal of this review was to emphasize the importance of understanding the materials and chemical properties of the various formulations to help build robustness into lubricated mechanisms. This can be effected in several ways. First and most important, the solid lubricant/antiwear coating formulation should be chosen to match the materials properties of the coated part, the expected Hertzian contact stress ($S_m$), the type and duration of relative contact motion, and the environment (vibration during launch, temperature, and gaseous, both before launch and in space). Next, prior to deposition of a coating, the part's surface can be appropriately hardened using a surface treatment or by deposition of a hard coating. In addition, the two base materials of the contact should be chemically dissimilar, so that if the worst happens and the lubricant and hard coatings are breached, the possibility of galling can be reduced. Finally, adequate testing of the lubricated part should be conducted, duplicating flight conditions as closely as possible (TLYF).

Solid-based tribology of spacecraft is a relatively mature field: most materials have been in use for three to five decades, and the various formulations used in space are well known. There are opportunities to develop more novel approaches that can improve performance and robustness, but the

spacecraft industry is highly conservative and concerned about "catastrophic improvement," where the cure is worse than the problem. HH-DLC coatings provide both extremely low friction and low wear in vacuum. However, there is no terrestrial market for these coatings like there is for more typical DLC coatings, and so are not being manufactured outside of laboratories. One way robustness can be built in to mechanisms that experience changing environments (temperature and gaseous) is by using adaptive "chameleon" materials/coatings, which are composites whose various components each provide excellent performance in different conditions. Surface engineering techniques like laser microtexturing of surfaces can provide enhanced cycle life by forming lubricant reservoirs. Finally, hybrid approaches using low-friction coatings together with greases or oils are beginning to be used in niche applications.

**Funding:** This research received no external funding.

**Acknowledgments:** The author gratefully acknowledges the collaboration and support of many coworkers from The Aerospace Corporation during the author's previous employment there.

**Conflicts of Interest:** The author declares no conflict of interest.

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
