# Peer review of "Effective Application of Solid Lubricants in Spacecraft Mechanisms"

_lubricants, doi:10.3390/lubricants8070074_

Round 1

Reviewer 1 Report

This is a comprehensive review on the selected field of engineering tribology in space environment. As stated in the abstract, it will be a very useful tool for designers and engineers in selecting solid lubricant material. However, the submission is a review paper instead of a handbook purely for material selection. I would like the author to provide the following sections.

  1. Latest research findings on the fundamental wear mechanisms and friction mechanisms of solid lubricants and other related materials, e.g. micro/nano/atomic observations of wear debris formation, tribofilm, tribo-chemistry (oxidation) and adhesive transfer of tribofilm.
  2. A record of laboratory tribological studies on the friction and wear properties of typical candidate lubricant coatings under selected environment conditions, e.g. elevated temperature, vacuum and in-air, as well as varied humidity. 
  3. The currently selected literature failed to reflect the state of the art development of tribological coatings. In addition to MoS2 and DLC coatings, other types of sputtered coatings have been widely used in wear protection and friction reduction, such as nanocomposites and multilayer hard coatings containing self-adaptive lubricants like CrN, VN. If these materials have not been used in the particular area, perhaps some comparative experimental study of the tribological properties would help guide the users in their material selection. 
  4. Finally, the title of the paper 'solid lubrication' or 'solid lubricants'? It seems the author described more about the latter. Please check.
  5. Section 5.3 was written as general statements instead of review because of lack of cited references.

Reviewer 2 Report

The manuscript examines various solid lubricants and tribological coatings used and potentially applicable in the aerospace industry.

In general, the manuscript is well organized and is of undoubted interest to the potential reader. Since the volume of the manuscript is large, in case the author does not begin to divide it into two parts, I recommend giving some content / structure for a better orientation of the reader.

I recommend to seriously change the abstract. Now it is more like another Introduction ("Solid lubrication ... have been important" - more suitable for Introduction). I recommend in the Abstract to give a brief description of the manuscript so that the reader can understand how much he is interested in the manuscript.

53 pages is too large for one article. Perhaps it makes sense to divide the manuscript into two parts.

Some general classification of currently available solid lubricants / tribological coatings would be useful (to some extent this is done in Table 2, but in my opinion this table is not complete - I would recommend, for example, also indicate hardness and elastic modulus).

While the author undoubtedly gave a useful comparison of solid and liquid lubricants (Table 1), as well as a comparison of the properties of various solid lubricants (Table 2), however, Table 2 does not have some of the elements considered - for example, DLC and PVD coatings.

Line 1476 - The DLC has already been decrypted previously (line 1222) - it makes no sense to re-decrypt this abbreviation.

The use of PVD coatings is considered in only one small paragraph (lines 850-855) - while there is great potential for the use of new generation PVD coatings (including multicomponent, nanolayer and nanostructured coatings, including MoS2, as well as molybdenum, chromium, yttrium oxides and hafnium, which have good tribological properties). An important advantage of PVD coatings is their perfect adhesion to the substrate, extremely high hardness and wear resistance combined with good tribological properties. Perhaps it would be worth spending a little more than five lines on them (out of 1600 lines). Only three coatings are named — TiC, TiN, and TiCN — all of which belong to the “first generation” of PVD coatings and are significantly inferior to coatings of new generations.

The author examines HH-DLC coatings in sufficient detail, but does not mention coatings based on cubic boron nitride cBN, which also have extremely high hardness and wear resistance combined with good tribological properties. Moreover, the heat resistance of cBN coatings is higher compared to HH-DLC.

Round 2

Reviewer 2 Report

This review is of undoubted interest and can be recommended for publication. The author took into account all the recommendations of the reviewer, making a number of useful changes.